# SYMMETRY-DRIVEN GRAPH NEURAL NETWORKS

## ABSTRACT

Exploiting symmetries and invariance in data is a powerful, yet not fully exploited, way to achieve better generalisation with more efficiency. In this paper, we introduce two graph network architectures that are equivariant to several types of transformations affecting the node coordinates. First, we build equivariance to any transformation in the coordinate embeddings that preserves the distance between neighbouring nodes, allowing for equivariance to the Euclidean group. Then, we introduce angle attributes to build equivariance to any angle preserving transformation - thus, to the conformal group. Thanks to their equivariance properties, the proposed models can be vastly more data efficient with respect to classical graph architectures, intrinsically equipped with a better inductive bias and better at generalising. We demonstrate these capabilities on a synthetic dataset composed of $n$-dimensional geometric objects. Additionally, we provide examples of their limitations when (the right) symmetries are not present in the data.

## 1 INTRODUCTION

Symmetries exist throughout nature. All the fundamental laws of physics are built upon the framework of symmetries, from the gauge groups describing the Standard Model of particle physics, to Einstein's theories of general and special relativity. Once one understands the symmetry of a certain system, powerful predictions can be made. A notable example is that of Gell-Mann's eightfold-way (Gell-Mann, 1961), built upon the symmetries observed in hadrons, that led to his prediction of the $\Omega^-$ baryon, which was subsequently observed 3 years later (Barnes et al., 1964). The study of symmetries and invariance in deep learning has recently become a field of interest to the community (see, *e.g.*, (Bronstein et al., 2021) for a comprehensive overview), and rapid progress has been made in constructing architectures with group theoretic structures embedded within. Two fundamental architectures in machine learning, the convolutional and graph neural networks, are invariant to the translation and permutation groups respectively.

Graph networks in particular are designed to learn from graph-structured data and are by construction invariant to permutations of the input nodes. They were originally proposed in (Gori et al., 2005; Scarselli et al., 2008) and have received a lot of attention in the last years (see, *e.g.*, (Battaglia et al., 2018; Hamilton, 2020; Wu et al., 2020) for a comprehensive overview). Due to their properties, they find application in a broad range of problems like learning the dynamics of complex physical systems (Sanchez-Gonzalez et al., 2020; Pfaff et al., 2021), particle identification in particle physics (Dreyer & Qu, 2021), learning causal and relational graphs (Kipf et al., 2018; Li et al., 2020), discovering symbolic models (Cranmer et al., 2020), as well as quantum chemistry (Gilmer et al., 2017) and drug discovery (Stokes et al., 2020).

In this work, we decouple node coordinates from other node attributes to obtain invariance (and possibly equivariance) to many transformations affecting the node coordinates, possibly belonging to important groups, in addition to permutation invariance. First, we define the *distance preserving graph network* (DGN) whose updated node, edge and global attributes are invariant to any transformation in the coordinate embeddings that preserves the distance between neighbouring nodes, while node coordinates can be updated in an equivariant way. Examples of such transformations are rotations and translations, but also transformations on the Hoberman sphere, whose 3D shape changes while preserving distances between connected nodes. Equivariance to dilations in the coordinates can be obtained via the conformal orthogonal group with the inclusion of an additional input layer. Then, we define the *angle preserving graph network* (AGN), whose updated node, edge and global attributes (resp. node coordinates) are invariant (resp. equivariant) to any transformation preserving

the angles between triples of neighbouring nodes in the graph (a notable example being molecular conformations) and, thus, to the $n$-dimensional conformal group on $\mathbb{R}^n$.

By constructing such architectures, we enable a wide range of possible transformations to be performed on graph-structured data, with the only requirement being that distances or angles between coordinates of neighbouring nodes are preserved. This means that the updated attributes/coordinates of both networks are invariant/equivariant to translations, rotations, reflections (*i.e.*, the Euclidean group, $E(n)$), with the AGN allowing also invariance/equivariance to dilations, inversions and non-orthogonal rotations. In practice, this means the networks are able to filter out many copies of the same input whose coordinate embeddings have been transformed and therefore learn more efficiently than architectures which consider the inputs as distinct. In other words, a single sample contains the same information as many copies of it obtained by appropriately transforming it.

Finally, while the two architectures we present are *partially* overlapping in terms of symmetries, it is important to consider specific use cases where the use of one architecture over another would be preferred. We test the architectures on a synthetic dataset consisting of $n$-dimensional geometric shapes and other benchmark datasets. Moreover, we show examples of cases in which using our architectures is counterproductive due to the lack of symmetries in the data.

## 2 BACKGROUND

### 2.1 GROUP THEORY AND EQUIVARIANCE

Group theory provides the mathematical formulation for symmetries of systems, with symmetry operations represented by individual group elements. A group can be defined as a set $G$ equipped with a binary operation, which enables one to combine two group elements to form a third, whilst preserving the group axioms (associativity, identity, closure and inverse). A function unaffected by a group action is said to be *invariant*, which is particular instance of *equivariance*. Formally, let $X \subseteq \mathbb{R}^n$ and $\varphi_g : X \to X$ be a transformation on $X$ for a group element $g \in G$. Then, the linear map $\Phi : X \to Y, Y \subseteq \mathbb{R}^n$, is equivariant to $G$ if $\forall g \in G, \exists \varphi'_g : Y \to Y$ such that

$$\Phi(\varphi_g(\mathbf{x})) = \varphi'_g(\Phi(\mathbf{x})), \quad \forall \varphi_g : X \to X .$$

When $\varphi'_g$ is the identity, we say that $\Phi$ is invariant to $G$. The functional form of $\Phi$ is determined by the specific group of interest. It is worth pointing out that, whilst not often expressed in group theoretic notation, equivariance exists in common deep learning architectures; the convolution operation used in CNNs is equivariant under the translation group and approximate invariance is typically achieved via pooling operations. Graph neural networks are invariant to the permutation group.

**Euclidean group** The translation and orthogonal groups ($T(n)$ and $O(n)$ respectively) act through translations, orthogonal rotations and reflections. The semidirect product of these two groups $E(n) = T(n) \rtimes O(n)$ is known as the Euclidean group, which consists of transformations that preserves distances. A function that is invariant under $E(n)$ is $\|\mathbf{x}_i - \mathbf{x}_j\|_2^2$, for any $\mathbf{x}_i, \mathbf{x}_j \in \mathbb{E}^n$ (see Appendix A for a proof).

**Conformal group** The conformal group $\text{Conf}(\mathbb{R}^{n,0})$ is the group of transformations $\varphi : \mathbb{R}^n \to \mathbb{R}^n$ that preserve angles. Formally, for any triple of vectors $\mathbf{x}_j, \mathbf{x}_i, \mathbf{x}_k \in \mathbb{R}^n$, let us denote by $\angle(\mathbf{x}_j, \mathbf{x}_i, \mathbf{x}_k)$ the angle centred on $\mathbf{x}_i$ with rays given by $\mathbf{x}_j - \mathbf{x}_i$ and $\mathbf{x}_k - \mathbf{x}_i$. Then, a conformal transformation $\varphi$ satisfies

$$\angle(\mathbf{x}_j, \mathbf{x}_i, \mathbf{x}_k) \to \angle(\varphi(\mathbf{x}_j), \varphi(\mathbf{x}_i), \varphi(\mathbf{x}_k)) = \angle(\mathbf{x}_j, \mathbf{x}_i, \mathbf{x}_k) . \tag{1}$$

By definition, the transformations of the conformal group include translations, rotations, reflections (which collectively form the Euclidean group $E(n)$, as well as dilations, inversions and other features. An important subgroup is the conformal orthogonal group, which requires transformations to be orthogonal (and thus, inversions are not allowed).

### 2.2 GRAPHS AND COORDINATE EMBEDDINGS

A graph is defined as $\mathcal{G}(\mathcal{V}, \mathcal{E})$ where $\mathcal{V} = \{1, \ldots, N\}$ is the set of nodes and $\mathcal{E} = \{(j, i)\} \subseteq \mathcal{V} \times \mathcal{V}$ is the set of (directed) edges connecting nodes in $\mathcal{V}$ (where $j$ and $i$ denote the source and target nodes

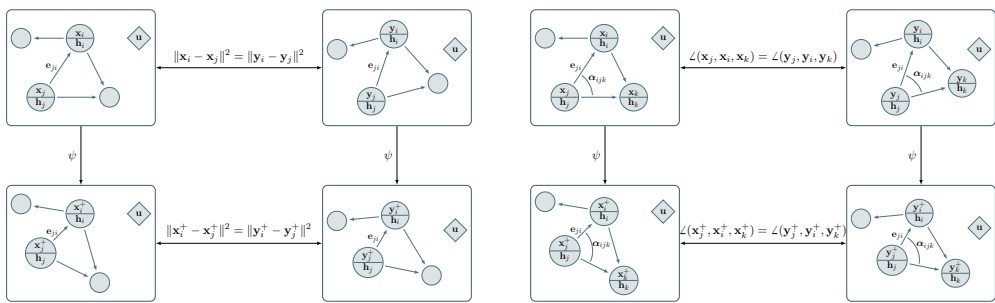

Figure 1: Representation of relative distance (left) and angle (right) preserving maps.

respectively). We define $\mathcal{N}_i \triangleq \{j \mid (j,i) \in \mathcal{E}\}$ as the set of (in-)neighbours of node $i$. Moreover, we associate node and edge features, $\mathbf{v}_i \in \mathbb{R}^{n_v}$, $\forall i \in \mathcal{V}$, and $\mathbf{e}_{ji} \in \mathbb{R}^{n_e}$, $\forall (j,i) \in \mathcal{E}$ respectively, to each node and edge in the graph as well as a global attribute $\mathbf{u} \in \mathbb{R}^{n_u}$. We assume that node features $\mathbf{v}_i$ consist of node coordinates $\mathbf{x}_i \in \mathbb{R}^{n_x}$ and additional features $\mathbf{h}_i \in \mathbb{R}^{n_h}$ (unrelated to coordinates), so that $\mathbf{v}_i = [\mathbf{x}_i, \mathbf{h}_i]$, $n_v = n_x + n_h$. With a slight abuse of notation, we denote $\mathcal{G}_X(\mathcal{V},\mathcal{E})$ and $\mathcal{G}_Y(\mathcal{V},\mathcal{E})$ as different coordinate embeddings of the same graph, meaning that they differ only in their node coordinates $\mathbf{x}_i$, while having the same node, edge and global attributes $\mathbf{h}_i$, $\mathbf{e}_{ji}$ and $\mathbf{u}$. Finally, let $\mathcal{A} \in \mathcal{V} \times \mathcal{V} \times \mathcal{V}$ be the set of (ordered) triples of nodes in a graph $\mathcal{G}$ that form an angle, i.e., $\mathcal{A} \triangleq \{(j,i,k) \mid j,k \in \mathcal{N}_i^u, j \neq k, \forall i \in \mathcal{V}\}$ with $\mathcal{N}_i^u \triangleq \{j \mid (j,i) \in \mathcal{E} \vee (i,j) \in \mathcal{E}\}$ being the set of in- and out-neighbors of node $i$.

We define *relative distance* and *angle preserving maps* as follows (see Figure 1).

**Definition.** *Let $\mathcal{G}_X(\mathcal{V},\mathcal{E})$ and $\mathcal{G}_Y(\mathcal{V},\mathcal{E})$ be such that $\|\mathbf{x}_i - \mathbf{x}_j\|^2 = \|\mathbf{y}_i - \mathbf{y}_j\|^2$, $\forall (i,j) \in \mathcal{E}$. Let $\mathbf{x}_i^+ = \psi(i, \mathcal{G}_X)$ and $\mathbf{y}_i^+ = \psi(i, \mathcal{G}_Y)$ for some function $\psi : \mathbb{R}^K \to \mathbb{R}^n$, $K \in \mathbb{Z}^+$. We say that $\psi$ is a* relative distance preserving map *if $\|\mathbf{x}_i^+ - \mathbf{x}_j^+\|^2 = \|\mathbf{y}_i^+ - \mathbf{y}_j^+\|^2$, $\forall (i,j) \in \mathcal{E}$.*

**Definition.** *Let $\mathcal{G}_X(\mathcal{V},\mathcal{E})$ and $\mathcal{G}_Y(\mathcal{V},\mathcal{E})$ be such that $\angle(\mathbf{x}_j, \mathbf{x}_i, \mathbf{x}_k) = \angle(\mathbf{y}_j, \mathbf{y}_i, \mathbf{y}_k) \; \forall(j,i,k) \in \mathcal{A}$. Let $\mathbf{x}_i^+ = \psi(i, \mathcal{G}_X)$ and $\mathbf{y}_i^+ = \psi(i, \mathcal{G}_Y)$ for some function $\psi : \mathbb{R}^K \to \mathbb{R}^n$, $K \in \mathbb{Z}^+$. We say that $\psi$ is a* relative angle preserving map *if $\angle(\mathbf{x}_j^+, \mathbf{x}_i^+, \mathbf{x}_k^+) = \angle(\mathbf{y}_j^+, \mathbf{y}_i^+, \mathbf{y}_k^+)$, $\forall(j,i,k) \in \mathcal{A}$.*

The simplest map satisfying the above definitions is the identity function

$$\mathbf{x}_i^+ = \psi(i, \mathcal{G}_X) = \mathbf{x}_i \tag{2}$$

which keeps the coordinates unchanged during the update step. Updating all the coordinates in the same way also trivially preserves both relative distances and angles. In the case where the coordinate embedding $X$ is a Conformal orthogonal transformation of $Y$ a possible map $\psi$ is defined by

$$\mathbf{x}_i^+ = \mathbf{x}_i + \sum_{j \in \mathcal{N}_i} a_{ji}(\mathbf{x}_j - \mathbf{x}_i). \tag{3}$$

with $a_{ji}$ possibly being a parametric function of other graph attributes. A thorough discussion and proofs are reported in in Appendix C.

## 3 EQUIVARIANT GRAPH NETWORKS

We start this section by recalling the structure of a standard graph network block. Then, we introduce two novel architectures with additional invariance and equivariance properties with respect to different node coordinate embeddings. We stress that while we build our discussion upon equivariance with respect to node coordinates (since there is, in general, no sense of equivariance of the network's node or edge embeddings), there may be cases in which our discussion can be extended to node or edge embeddings.

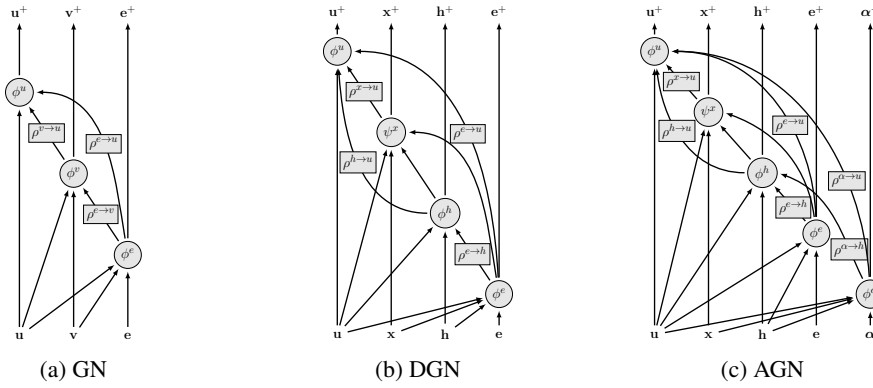

(a) GN        (b) DGN        (c) AGN

Figure 2: Graphical representation of the graph architectures

### 3.1 THE STARTING POINT - GRAPH NETWORK BLOCK (GN)

A general standard graph network block can be defined in terms of edge, node and global updates as

$$\mathbf{e}_{ji}^+ = \phi^e\big(\mathbf{v}_j, \mathbf{v}_i, \mathbf{e}_{ji}, \mathbf{u}\big), \qquad\qquad \forall (j,i) \in \mathcal{E} \qquad (4\text{a})$$

$$\mathbf{v}_i^+ = \phi^v\big(\mathbf{v}_i, \rho^{e\to v}(\{\mathbf{e}_{ji}^+\}_{j\in\mathcal{N}_i}), \mathbf{u}\big), \qquad\qquad \forall i \in \mathcal{V} \qquad (4\text{b})$$

$$\mathbf{u}^+ = \phi^u\big(\rho^{v\to u}(\{\mathbf{v}_i^+\}_{i\in\mathcal{V}}), \rho^{e\to u}(\{\mathbf{e}_{ji}^+\}_{(j,i)\in\mathcal{E}}), \mathbf{u}\big) \qquad\qquad (4\text{c})$$

where $\phi^e : \mathbb{R}^{2n_v+n_e+n_u} \to \mathbb{R}^{n_e^+}$, $\phi^v : \mathbb{R}^{n_v+n_e^++n_u} \to \mathbb{R}^{n_v^+}$, $\phi^u : \mathbb{R}^{n_v^++n_e^++n_u} \to \mathbb{R}^{n_u^+}$ are update functions (usually defined as neural networks whose parameters are to be learned) and $\rho^{e\to v}, \rho^{e\to u}, \rho^{v\to u}$ are aggregation functions reducing a set of elements of variable length to a single one via an input's permutation equivariant operation like element-wise summation or mean.

### 3.2 DISTANCE PRESERVING GRAPH NETWORK BLOCK (DGN)

By decoupling the updates of node coordinates and features ($\mathbf{x}_i$ and $\mathbf{h}_i$), the DGN block is defined through the following updates

$$\mathbf{e}_{ji}^+ = \phi^e\big(\mathbf{e}_{ji}, \mathbf{h}_i, \mathbf{h}_j, \|\mathbf{x}_i - \mathbf{x}_j\|_2^2, \mathbf{u}\big), \qquad\qquad \forall (j,i) \in \mathcal{E} \qquad (5\text{a})$$

$$\mathbf{h}_i^+ = \phi^h\big(\rho^{e\to h}(\{\mathbf{e}_{ji}^+\}_{j\in\mathcal{N}_i}), \mathbf{h}_i, \mathbf{u}\big), \qquad\qquad \forall i \in \mathcal{V} \qquad (5\text{b})$$

$$\mathbf{x}_i^+ = \psi^x\big(i, \mathcal{G}_X\big), \qquad\qquad \forall i \in \mathcal{V} \qquad (5\text{c})$$

$$\mathbf{u}^+ = \phi^u\big(\rho^{e\to u}(\{\mathbf{e}_{ji}^+\}_{(j,i)\in\mathcal{E}}), \rho^{h\to u}(\{\mathbf{h}_i^+\}_{i\in\mathcal{V}}), \rho^{x\to u}(\{\|\mathbf{x}_i^+ - \mathbf{x}_j^+\|_2^2\}_{(j,i)\in\mathcal{E}}), \mathbf{u}\big) \qquad (5\text{d})$$

where $\phi^e : \mathbb{R}^{n_e+2n_h+n_u+1} \to \mathbb{R}^{n_e^+}$, $\phi^h : \mathbb{R}^{n_e^++n_h+n_u} \to \mathbb{R}^{n_h^+}$, $\phi^u : \mathbb{R}^{n_e^++n_h^++n_u+1} \to \mathbb{R}^{n_u^+}$ are the update functions, $\rho^{e\to h}, \rho^{e\to u}, \rho^{h\to u}$ are aggregation functions and $\psi^x : \mathbb{R}^K \to \mathbb{R}^{n_x}, K \in \mathbb{Z}^+$ is some, possibly parametric, relative distance preserving map.

Because of the way in which node coordinates are processed to update edge and global embeddings (*i.e.*, only through their relative distances), it can be easily seen that $\mathbf{e}_{ji}^+$, $\mathbf{h}_i^+$ and $\mathbf{u}^+$ are, by construction, invariant to any transformation of the input coordinates that locally maintains their relative distances along the edges defined by the graph structure (a notable example being the one of Euclidean transformations; see Appendix D for a proof). Moreover, updated node coordinates $\mathbf{x}_i^+$ can be invariant or equivariant to (some of) those transformations, depending on the particular structure of $\psi^x$. For example, using equation 3 or the trivial identity function (equation 2) would result in coordinates being updated in an equivariant way with respect to Euclidean transformations, with $a_{ji}$ possibly being parametric functions (*e.g.*, $a_{ji} = \phi^x(\mathbf{e}_{ji})$).

### 3.3 ANGLE PRESERVING GRAPH NETWORK BLOCK (AGN)

Given a graph $\mathcal{G}(\mathcal{V}, \mathcal{E})$, let $\boldsymbol{\alpha}_{jik} \in \mathbb{R}^{n_\alpha}$ be the angle embedding associated to each angle $(j,i,k) \in \mathcal{A}$ and assume that $\boldsymbol{\alpha}_{jik}$ contain no information about the angles $\angle(\mathbf{x}_j, \mathbf{x}_i, \mathbf{x}_k)$. Moreover, define $\mathcal{A}_i$

as the set of (ordered) couples of nodes forming an angle whose vertex is node $i$, *i.e.*, $\mathcal{A}_i \triangleq \{(j,k) \mid (j,y,k) \in \mathcal{A}, y = i\}$. Then, the AGN block is then characterised by the following updates

$$\boldsymbol{\alpha}_{jik}^+ = \phi^\alpha(\mathbf{h}_i, \mathbf{h}_j, \mathbf{h}_k, \boldsymbol{\alpha}_{jik}, \angle(\mathbf{x}_j, \mathbf{x}_i, \mathbf{x}_k), \mathbf{u}), \qquad\qquad \forall (j,i,k) \in \mathcal{A} \quad \text{(6a)}$$

$$\mathbf{e}_{ji}^+ = \phi^e(\mathbf{h}_j, \mathbf{h}_i, \mathbf{e}_{ji}, \mathbf{u}), \qquad\qquad \forall (j,i) \in \mathcal{E} \quad \text{(6b)}$$

$$\mathbf{h}_i^+ = \phi^h(\mathbf{h}_i, \rho^{e \to h}(\{\mathbf{e}_{ji}^+\}_{j \in \mathcal{N}_i}), \rho^{\alpha \to h}(\{\boldsymbol{\alpha}_{jik}^+\}_{(j,k) \in \mathcal{A}_i}), \mathbf{u}), \qquad\qquad \forall i \in \mathcal{V} \quad \text{(6c)}$$

$$\mathbf{x}_i^+ = \psi^x(i, \mathcal{G}_X), \qquad\qquad \forall i \in \mathcal{V} \quad \text{(6d)}$$

$$\mathbf{u}^+ = \phi^u(\rho^{h \to u}(\{\mathbf{h}_i^+\}_{i \in \mathcal{V}}), \rho^{e \to u}(\{\mathbf{e}_{ji}^+\}_{(j,i) \in \mathcal{E}}), \rho^{\alpha \to u}(\{\boldsymbol{\alpha}_{jik}^+\}_{(j,i,k) \in \mathcal{A}}), \mathbf{u}) \qquad\qquad \text{(6e)}$$

where $\phi^\alpha, \phi^e, \phi^h, \psi^x, \phi^u$ are the update functions[1], and $\rho^{e \to h}, \rho^{\alpha \to h}, \rho^{h \to u}, \rho^{e \to u}, \rho^{\alpha \to u}$ are the aggregation functions, with $\psi^x$ being a (possibly parametric) *relative angle preserving map*. Variants of this network that still preserve its properties can be easily constructed by, *e.g.*, using edge embeddings to update the angle embeddings, using angles to update edges or not considering angle attributes at all. Some examples are reported in Appendix F.1. It can be easily seen that $\boldsymbol{\alpha}_{jik}^+, \mathbf{e}_{ji}^+, \mathbf{h}_i^+, \mathbf{u}^+$ are invariant to any transformation of the coordinate embeddings that preserves the angles created by neighbouring nodes in the graph, while updated coordinates $\mathbf{x}_i^+$ can be invariant or equivariant to (some of) the same transformations, depending on the structure of $\psi^x$. As with the DGN, equivariance to the conformal orthogonal group can be achieved by using equation 3. A particular case of transformations the AGN can deal with is the one of conformal transfomations (see Appendix E for a formal discussion).

## 4 DISCUSSION

In the previous section we introduced two novel graph architectures. To construct them we have mainly built upon the actions on a coordinate system of two transformations that preserves distances and angles between neighbouring nodes in a graph. These transformations include as particular cases those in the Euclidean and conformal groups. Equivariance under $E(n)$ means equivariance to orthogonal rotations and translations, while by considering the full conformal group, we can further generalise our neural network architecture. Conformal $n$-dimensional transformations consist of the groups containing translations, dilations, rotations and inversions with respect to an $n-1$ sphere. A conformal transformation is therefore a powerful tool for mapping data points onto each other, and hence, building a neural network architecture invariant to the conformal group enables the architecture to be invariant to a wide selection of interesting subgroups. By introducing distance and angle preserving transformations that take into account the structure of the graph, invariance and equivariance under more general transformations have been achieved. A notable example is that the DGN architecture, whose updates are invariant to any transformation of a Hoberman sphere, while both the DGN and AGN can be invariant to transformations affecting only subsets of their nodes, as shown below.

In summary, by taking advantage of the powerful tools of group theory, and performing operations on a vector space which are invariant or equivariant under group transformations, we can build a neural network architecture which can take advantage of these group properties. Such an architecture will be able to deal with rotated, translated, dilated (or more generally transformed) data more efficiently than a standard graph network which does not have the above group properties built into it.

### 4.1 BEYOND GLOBAL TRANSFORMATIONS

So far, we have talked about global transformations in the coordinate space. In general, a global symmetry $\varphi_g$, as defined in Section 2.1 acts on a function $\psi(\mathbf{x})$ as

$$\psi(\mathbf{x}) \to \varphi_g \psi(\mathbf{x}). \qquad\qquad (7)$$

A *local* group symmetry $\varphi_l(\mathbf{x})$ is instead defined as

$$\psi(\mathbf{x}) \to \varphi_l(\mathbf{x})\psi(\mathbf{x}), \qquad\qquad (8)$$

---

[1]The update functions in equation 6 live in the function spaces $\phi^\alpha : \mathbb{R}^{3n_h + n_\alpha + 1 + n_u} \to \mathbb{R}^{n_\alpha^+}$, $\phi^e : \mathbb{R}^{2n_h + n_e + n_u} \to \mathbb{R}^{n_e^+}$, $\phi^h : \mathbb{R}^{n_h + n_e^+ + n_\alpha^+ + n_u} \to \mathbb{R}^{n_h^+}$, $\psi^x : \mathbb{R}^K \to \mathbb{R}^{n_x}$, for an appropriate $K \in \mathbb{Z}^+$ depending on the actual number of parameters, and $\phi^u : \mathbb{R}^{n_h^+ + n_e^+ + n_\alpha^+ + n_u} \to \mathbb{R}^{n_u^+}$.

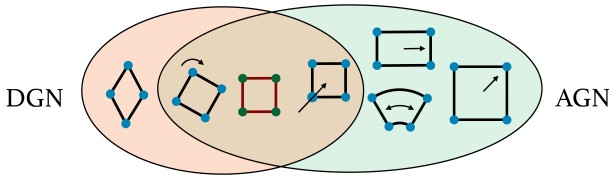

Figure 3: Equivariance of the two networks. The green/red square denotes the base graph and the blue/black ones alternative coordinate embeddings, equivariant for the DGN (red) or AGN (green).

where now the group action can differ at all points.

The architectures we present are also able to deal with some local symmetries. Namely, the DGN and AGN preserve local distances and angles respectively *where a suitable subgraph must be defined*. As an example consider a transformation that rotates only some nodes of a graph; all nodes which are in the neighbourhoods of the nodes will, in the angular architecture, have their angles preserved, and in the distance-preserving architecture, the distances between nodes will be preserved. A pertinent use-case for such a symmetry is that of molecular conformations, where the spatial arrangement of the atoms can be interconverted by rotations about formally single bonds). Both the DGN and the AGN updates would be invariant to such a transformation at the level of the two distinct subgraphs separated by the bond in question, thus enabling one to learn conformation-invariant properties.

### 4.2 COMPARING AND EXTENDING THE TWO ARCHITECTURES

As mentioned, the two architectures we presented are partially overlapping in terms of equivariance features. Whilst both of them contain invariance to $E(n)$, the DGN can deal with non-orthogonal transformations that preserve the distance between neighbouring nodes (such as with the Hoberman sphere) while the AGN is invariant to the conformal group which includes non-orthogonal transformations that preserve angles but not distances (see Figure 3).

The DGN can be equipped with a sense of scale invariance through an input scaling layer that makes it invariant to the conformal orthogonal group. Let $\gamma = \alpha / \max_{(i,j) \in \mathcal{E}} \|\mathbf{x}_i - \mathbf{x}_j\|$ for some $\alpha \in \mathbb{R}$, where we identify $\gamma$ as satisfying the dilation condition required under the conformal orthogonal group. Then, conformal orthogonal invariance can simply be obtained by computing scale-normalised coordinates $\tilde{\mathbf{x}}_i = \gamma \mathbf{x}_i$ and using them as the coordinates in equation 5.

Introducing coordinate scaling therefore enables the DGN to be scale invariant, under the condition that all transformations remain orthogonal. With the AGN, we can relax this condition and be invariant under the full conformal group. This larger group invariance enables us to work with very powerful transformations on data, which can also be dangerous. For example, if the scale is an important property of the dataset, the AGN will not recognise it and so will learn incorrect information about the data. Similarly, one must also not use the DGN blindly and irrespective of the properties of the dataset; as the architecture is invariant to transformations that preserve distance between neighbouring nodes but not angles it is able to, for example, map a square onto a line. The two architectures we presented can be also combined together at the price of losing some features, while generalising to both distance and angle preserving transformations.

## 5 RELATED WORK

The study and formulation of group equivariant neural networks have flourished in the last years and they have proven to be extremely powerful in many tasks. The first example is probably that of convolutional neural networks (LeCun et al., 1990), which exploits translation equivariance and invariance, thanks to the convolution and pooling operations respectively, and have led to breakthroughs in most vision tasks. CNNs have been generalised to exploit larger groups of symmetries. G-CNNs are proposed in (Cohen & Welling, 2016) to deal with more general symmetries, including rotations and translations, and extended in (Bekkers, 2020; Finzi et al., 2020) to deal with Lie groups. Equivariance to arbitrary symmetry groups can be achieved via self-attention mechanisms in (Romero & Cordonnier, 2021). Continuous convolutions are used in SchNet (Schütt et al., 2017)

to achieve $E(n)$ invariance, and $SE(3)$ equivariance is achieved in Thomas et al. (2018) via the use of spherical harmonics. The drawback of many of these methods is their limited applicability due to computational complexity.

There has been some work on constructing general MLPs that are equivariant to different groups; a layer equivariant to general matrix groups is presented in Finzi et al. (2021), whilst equivariance to the Lorentz group for physics applications has recently been explored (Bogatskiy et al., 2020). Further applications include (Mattheakis et al., 2019), where physical symmetries are embedded in neural networks via embedding physical constraints in the structure of the network, and in (Barenboim et al., 2021) where the ability of neural networks to discover and learn symmetries in data is explored.

Graph neural networks are, by construction, permutation invariant (Scarselli et al., 2008; Battaglia et al., 2018). Recently, there has been a lot of work on building equivariance to other interesting groups in GNNs. There has been particular interest in the Euclidean group with results for the subgroups $SE(3)$ and $E(3)$ obtained in (Thomas et al., 2018; Fuchs et al., 2020; Köhler et al., 2020; Finzi et al., 2020; Batzner et al., 2021; Yang et al., 2020). Extending these architectures, a message-passing convolutional graph layer is proposed in (Satorras et al., 2021) which implements $E(n)$ invariance for edge and node features and equivariance for node coordinates, leading to state of the art results in a variety of tasks, including $n$-body particle systems simulations and molecular property predictions. A similar idea has been also proposed in (Farina & Slade, 2021), where a general $E(n)$ equivariant GNN is presented, and in (Horie et al., 2021), in which equivariance to isometric transformations is considered. These $E(n)$ equivariant networks can be seen as special instances of our DGN architecture, where particular aggregation functions are applied and a particular choice is made for the distance preserving map $\psi^x$ (see Appendix G). In (Satorras et al., 2021), for example, the map $\psi^x$ is chosen so that relative distances between all possible pairs of node in the graph are preserved, thus allowing only for rigid transformations in $E(n)$.

Angular information is used in (Smith et al., 2017) but no attention is explicitly paid to equivariance. DimeNET is proposed in (Klicpera et al., 2020), where both distance and angle embeddings computed through embeddings in novel orthogonal basis functions are used in a message passing graph network to achieve equivariance. State of the art results are obtained on molecular property and dynamics prediction datasets. We note, however, that while DimeNET produces state of the art results, it is restricted to atomic data, due to the requirement of Gaussian radial and Bessel functions as the basis representations. Additionally, angular embeddings computed in the isometric invariant layer in (Horie et al., 2021) corresponds to the extraction of both relative distances and angles of each pair of vertices. Again, these approaches are special instances of our networks where particular aggregation and learnable functions or hand engineered distance and/or angle embeddings are used (see Appendix G). In this sense, our network is unconstrained and can learn what it needs. Moreover, thanks to its general form it can achieve invariance to the conformal group (and more) which, to the best of our knowledge, does not hold for known architectures.

In summary, most of the existing equivariant graph architectures are at most equivariant to the Euclidean group (represented as the intersection set in Figure 3), whilst the architectures we propose allow, in their general form, more general transformations.

## 6 EXPERIMENTAL RESULTS

In this section we give an insight into the performance of the architectures we proposed and on their limitations. First, we demonstrate that perfect generalisation to unseen data can be achieved on datasets with a large number of symmetries, possibly accompanied by a faster convergence rate (as shown for QM9 in Appendix H.4). Then, we show that, when used on datasets with few symmetries, using our architectures can be counterproductive. In general, when the right symmetries are present, equivariant architectures experience both increased accuracy and convergence speed, as demonstrated by state of the art results obtained on a number of tasks by architectures that can be obtained as particular instances of the ones proposed here. Examples include $n$-body system dynamics prediction, molecular dynamics and molecular property prediction tasks (Satorras et al., 2021; Klicpera et al., 2020; Horie et al., 2021).

## 6.1 POLYTOPES CLASSIFICATION

We consider a $n$-dimensional polytopes classification problems for $n = 3, 4, 5$. The datasets are composed of graph representations of regular polytopes and the number of classes varies with $n$. Simplexes, hypercubes and orthoplexes are considered for all values of $n$. For $n = 3$ also dodecahedra and icosahedra are added, while for $n = 4$ we consider $24-$, $120-$ and $600-$ cell polytopes. Additional results are reported in Appendix H.2. The training dataset consists of a single graph per polytope where the graph is specified in terms of node coordinates $\mathbf{x}_i$ (so that the resulting polytope is inscribed in a hypersphere of radius 1) and list of edges $\mathcal{E}$ (and possibly angles $\mathcal{A}$); node, edge and possible angle attributes $(\mathbf{h}_i, \mathbf{e}_{ji}, \boldsymbol{\alpha}_{jik})$ are fixed to be equal to 0 for every node in the graph. This makes the problem intrinsically harder since only coordinates can be used to discern between polytopes. The test set is composed of randomly transformed versions of those in the training set, whose coordinates are obtained as $\tilde{\mathbf{x}}_i = \gamma A \mathbf{x}_i + q$ for some $\gamma \in \mathbb{R}$, $A \in \mathbb{R}^{n \times n}$ and $q \in \mathbb{R}^n$. Note that, while even a standard GN could learn to correctly classify transformed polytopes, a much larger training dataset is needed in order to do so, as shown in Appendix H.

We compare graph networks built with AGN, DGN and standard (GN) blocks, as well as EGNN (Satorras et al., 2021), DimeNET (Klicpera et al., 2020) and SE3-Transformer (Fuchs et al., 2020), the latter only for $n = 3$. The DGN block is also combined with a scaling layer (SDGN) as in Section 4.2 to obtain equivariance to dilations. Architectures based on AGN, DGN and GN blocks employ sum aggregation functions $\rho$ and $\psi^x$ set to equation 3 (other cases are considered in the appendix). Results are reported in Table 1.

| | | | AGN | SDGN | DGN | GN | SE3-Trans | DimeNET | EGNN |
|---|---|---|---|---|---|---|---|---|---|
| | | training accuracy | **1** | **1** | **1** | **1** | **1** | **1** | **1** |
| $n = 3$ | test accuracy | Orthogonal | 1 | 1 | 1 | 0.44 ± 0.15 | 1 | 1 | 1 |
| | | Orthogonal + dilation | 1 | 1 | 0.45 ± 0.05 | 0.46 ± 0.14 | 1 | 0.36 ± 0.06 | 0.36 ± 0.02 |
| | | Non-orthogonal ($\mu = 0.5$) | 1 | 1 | 0.44 ± 0.12 | 0.47 ± 0.15 | 1 | 0.34 ± 0.08 | 0.33 ± 0.08 |
| | | Non-orthogonal ($\mu = 1.5$) | 1 | **0.93 ± 0.07** | 0.41 ± 0.07 | 0.44 ± 0.17 | 1 | 0.33 ± 0.10 | 0.33 ± 0.12 |
| | | Non-orthogonal ($\mu = 3.0$) | 1 | 0.83 ± 0.15 | 0.37 ± 0.04 | 0.43 ± 0.15 | 1 | 0.31 ± 0.03 | 0.31 ± 0.07 |
| | | training accuracy | **1** | **1** | **1** | **1** | – | **1** | **1** |
| $n = 4$ | test accuracy | Orthogonal | 1 | 1 | 1 | 0.53 ± 0.04 | – | 1 | 1 |
| | | Orthogonal + dilation | 1 | 1 | 0.61 ± 0.05 | 0.51 ± 0.03 | – | 0.52 ± 0.06 | 0.55 ± 0.07 |
| | | Non-orthogonal ($\mu = 0.5$) | 1 | 0.96 ± 0.03 | 0.60 ± 0.08 | 0.48 ± 0.12 | – | 0.50 ± 0.07 | 0.56 ± 0.06 |
| | | Non-orthogonal ($\mu = 1.5$) | 1 | 0.83 ± 0.12 | 0.61 ± 0.06 | 0.49 ± 0.06 | – | 0.51 ± 0.08 | 0.54 ± 0.07 |
| | | Non-orthogonal ($\mu = 3.0$) | 1 | 0.77 ± 0.17 | 0.59 ± 0.08 | 0.51 ± 0.04 | – | 0.53 ± 0.03 | 0.53 ± 0.11 |
| | | training accuracy | **1** | **1** | **1** | **1** | – | **1** | **1** |
| $n = 5$ | test accuracy | Orthogonal | 1 | 1 | 1 | 0.64 ± 0.07 | – | 1 | 1 |
| | | Orthogonal + dilation | 1 | 1 | 0.46 ± 0.05 | 0.55 ± 0.04 | – | 0.58 ± 0.08 | 0.55 ± 0.06 |
| | | Non-orthogonal ($\mu = 0.5$) | 1 | **0.99 ± 0.01** | 0.41 ± 0.05 | 0.60 ± 0.13 | – | 0.44 ± 0.03 | 0.41 ± 0.04 |
| | | Non-orthogonal ($\mu = 1.5$) | 1 | **0.98 ± 0.02** | 0.39 ± 0.08 | 0.61 ± 0.16 | – | 0.45 ± 0.11 | 0.44 ± 0.03 |
| | | Non-orthogonal ($\mu = 3.0$) | 1 | **0.98 ± 0.02** | 0.41 ± 0.10 | 0.58 ± 0.08 | – | 0.42 ± 0.06 | 0.43 ± 0.04 |

Table 1: Polytopes classification: training and test accuracy (mean ± standard deviation over 10 runs) for different transformations in the test set.

We note that all models reach a perfect accuracy on the training set. As for the test accuracy, results vary across the different models, in line with what one expects from their invariance properties. The standard GN cannot generalise to any type of transformation. When only orthogonal transformations are performed in the test set (*i.e.*, $A^\top A = I$, $\gamma = 1$) all models (except GN) perfectly generalise to unseen data. However, as expected, when also adding dilations (*i.e.*, $A^\top A = \gamma I$, $\gamma \in \mathbb{R}$), only AGN and SDGN (and the SE3-Transformer for $n = 3$) can generalise well. Finally, when adding random non-orthogonal transformations (*i.e.*, $\mu = \mathrm{E}[\|A^\top A - I\|_F] > 0$, $\gamma \in \mathbb{R}$), something possibly unexpected occurs. The AGN (and partially also the SDGN and the SE3-Transformer for $n = 3$) perfectly generalises to unseen polytopes whose neither angles nor edge lengths have been preserved (due to the non-orthogonality of $A$). This is probably due to the networks having learnt to distinguish polytopes based on the sum (or mean) of their angles which is preserved under non-orthogonal transformations due to the Gram–Euler theorem. While this behaviour turns out to be useful in the task at hand, one can easily envisage cases in which it may cause issues.

## 6.2 BENCHMARK DATASETS

Now we consider benchmark datasets from (Dwivedi et al., 2020) for which node coordinates are provided. In particular, we consider MNIST, CIFAR10 as graph classification problems and TSP as an edge classification one. When treated as images, MNIST and CIFAR10 are usually examples of datasets containing symmetries. However, when represented as graphs as in (Dwivedi et al., 2020),

those symmetries are mostly lost. This is due to, for example, the fact that both the foreground and the background of the images are embedded in the graphs. Furthermore, in TSP there are no apparent symmetries. In these cases, as one may expect, using an equivariant network would not produce any benefits. Rather, as we show, negative effects can appear.

We trained our models and a standard GNN on these datasets. All models have roughly the same number of parameters and the results are reported in Table 2. As can be seen, treating node coordinates as if they can posses some symmetric between among different samples turns out to be counterproductive both in terms of convergence speed and accuracy. In particular we see that while the standard GN has the best performance, the DGN is slightly better than the AGN. This is probably due to the fact that, while some non-orthogonal transformations mostly preserving local distances may be present in the dataset, angle preserving ones are much more rare. On TSP in particular, using angle related properties results in a significant drop in performance.

| block | MNIST | | CIFAR10 | | TSP | |
|---|---|---|---|---|---|---|
| | train acc | test acc | train acc | test acc | train F1 (positive) | test F1 (positive) |
| AGN | $0.943 \pm 0.004$ | $0.932 \pm 0.004$ | $0.644 \pm 0.004$ | $0.590 \pm 0.004$ | $0.632 \pm 0.011$ | $0.681 \pm 0.020$ |
| DGN | $0.957 \pm 0.001$ | $0.945 \pm 0.006$ | $0.657 \pm 0.004$ | $0.592 \pm 0.002$ | $0.748 \pm 0.013$ | $0.771 \pm 0.014$ |
| GN | $\mathbf{0.982 \pm 0.001}$ | $\mathbf{0.977 \pm 0.002}$ | $\mathbf{0.719 \pm 0.007}$ | $\mathbf{0.657 \pm 0.001}$ | $\mathbf{0.772 \pm 0.041}$ | $\mathbf{0.793 \pm 0.033}$ |

Table 2: Benchmark datasets. Train and test accuracy are reported for MNIST and CIFAR10. For TSP, due to the high class unbalance, the train and test F1 score for the positive class is reported.

### 6.2.1 COMPUTATIONAL COMPLEXITY

All the experiments have been run on an NVIDIA V100 GPU. The average time required for each training step (*i.e.*, forward pass, back-propagation and optimiser step for a batch of data) is reported in Table 3 for the architectures presented in this paper, normalised with respect to the batch-size. For the polytopes classification experiment we report the results obtained by using equation 2 and equation 3 (in parentheses). Using the scaling layer for the DGN drastically increase the computational complexity. Moreover, it can be seen that while the AGN and DGN have similar complexities for dataset in which the number of edges (and hence angles) is relatively low, when graphs are (nearly) complete (like in TSP) or highly connected, the AGN has an overhead with respect to the DGN (due to the need of updated and computing angles in addition to edges).

| | Time per training step (ms) | | | | | | |
|---|---|---|---|---|---|---|---|
| | polytopes (n=3) | polytopes (n=4) | polytopes (n=5) | MNIST | CIFAR10 | TSP | QM9 |
| batch-size | 5 | 6 | 3 | 128 | 128 | 8 | 512 |
| AGN | 3.3 (3.8) | 3 (3.8) | 5.6 (6.3) | 1.17 | 1.64 | 35 | 0.23 |
| SDGN | 4.4 (5) | 14.5 (15.2) | 7 (8) | 7.81 | 12.5 | - | - |
| DGN | 3.3 (3.8) | 3 (3.7) | 5.3 (6.3) | 0.78 | 1.09 | 16 | 0.19 |
| GN | 3.2 | 3 | 5.2 | 0.77 | 1.02 | 14 | 0.16 |

Table 3: Time per training step in the various experiments, normalised with respect to the batch-size.

## 7 CONCLUSION

In this paper we have presented novel deep learning architectures which are equivariant to distance and angle preserving transformations in graph coordinate embeddings. In particular, we have shown invariance or equivariance to the $E(n)$, $\mathrm{CO}(\mathbb{R}^n, \mathcal{Q})$ and $\mathrm{Conf}(\mathbb{R}^{n,0})$ groups in addition to permutation invariance. Invariance to local symmetries can also be achieved when the different transformations are applied to suitable subgraphs. We have applied our models to a synthetic dataset composed of $n$-dimensional regular polytopes as well as to several benchmark datasets. We have shown that the architectures we propose are significantly more accurate and data efficient than a standard graph network on datasets where there are large numbers of symmetries in the data. We also explicitly show examples where our architecture produces unexpected results or would not be applicable, due to a lack of symmetry in the data.

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

## A $\;\;E(n)$ INVARIANCE OF $\|\mathbf{x}_i - \mathbf{x}_j\|_2^2$

To show that $\|\mathbf{x}_i - \mathbf{x}_j\|_2^2$ is invariant under $E(n)$, it is sufficient to see that, under translation, one has

$$\mathbf{x} \to \mathbf{x}^+ = \mathbf{x} + \mathbf{z}, \quad \mathbf{z} \in \mathbb{E}^n \,,$$

while under rotation

$$\mathbf{x} \to \mathbf{x}^+ = Q\mathbf{x}, \quad Q \in O(n) \,.$$

Now, by using the above relations and the fact that $E(n)$ is the semidirect product of $T(n)$ and $O(n)$, it is easy to show that $\|\mathbf{x}_i - \mathbf{x}_j\|_2^2$ is invariant under $E(n)$ since

$$\|\mathbf{x}_i - \mathbf{x}_j\|_2^2 \to \|\mathbf{x}_i^+ - \mathbf{x}_j^+\|_2^2 = (Q(\mathbf{x}_i + \mathbf{z}) - Q(\mathbf{x}_j + \mathbf{z}))^\top (Q(\mathbf{x}_i + \mathbf{z}) - Q(\mathbf{x}_j + \mathbf{z})) \tag{9a}$$

$$= (Q\mathbf{x}_i - Q\mathbf{x}_j)^\top (Q\mathbf{x}_i - Q\mathbf{x}_j) \tag{9b}$$

$$= (\mathbf{x}_i - \mathbf{x}_j)^\top Q^\top Q (\mathbf{x}_i - \mathbf{x}_j) \tag{9c}$$

$$= (\mathbf{x}_i - \mathbf{x}_j)^\top (\mathbf{x}_i - \mathbf{x}_j) \,, \tag{9d}$$

where we used the fact that $Q^\top Q = I \; \forall Q \in O(n)$.

## B $\;\;$ INVARIANCE UNDER THE CONFORMAL ORTHOGONAL GROUP

In this section we will derive the result from Section 2.1 in the main text, that dilations and orthogonal rotations are transformations under $CO(\mathbb{R}^n, \mathcal{Q})$. The group is defined for a vector space $V$ with a quadratic form $\mathcal{Q}$. The group contains the linear transformations $\varphi : \mathcal{T} \to V$. The group action is defined as

$$\mathcal{Q}(\mathcal{T}x) = \gamma^2 \mathcal{Q}(x) \,, \tag{10}$$

where $\mathcal{T}$ is the set of linear transformations that we need to define and $\gamma$ is a scalar. For our purposes, we can consider a positive-definite quadratic form on $\mathbb{R}^n$; as we restrict our discussion to Euclidean space then the relevant quadratic form is

$$\mathcal{Q} = \sum_i x_i^2, \tag{11}$$

and so inserting equation 11 into equation 10 we have the condition

$$\sum_i (\mathcal{T}x_i)^2 = \gamma^2 \sum_i x_i^2.$$

If we consider, as above, an orthogonal transformation $Q$, and in addition, a dilation $x_i \to \gamma x_i$, then it is simple to show that, with $\mathcal{T} = \gamma Q$

$$\sum_i (\gamma Q x_i)^2 = \gamma^2 Q^\top Q \sum_i x_i^2 = \gamma^2 \sum_i x_i^2.$$

## C    RELATIVE DISTANCE AND ANGLE PRESERVING MAPS

Many different relative distance and/or angle preserving maps to be used in the coordinate update can be obtained in different ways, eventually requiring further assumptions on the data.

The simplest one is the identity function

$$\mathbf{x}_i^+ = \psi(i, \mathcal{G}_X) = \mathbf{x}_i \tag{12}$$

which keeps the coordinates unchanged during the update step. In this case both relative distances and angles are trivially preserved. Updating all the coordinates in the same way also trivially preserves both relative distances and angles. Examples include

$$\mathbf{x}_i^+ = a\mathbf{x}_i, \qquad\qquad a \in \mathbb{R}, \tag{13}$$
$$\mathbf{x}_i^+ = \mathbf{x}_i + \mathbf{a}, \qquad\qquad \mathbf{a} \in \mathbb{R}^{n_x} \tag{14}$$
$$\mathbf{x}_i^+ = Q\mathbf{x}_i + \mathbf{a}, \qquad\qquad Q \in \mathbb{R}^{n_x \times n_x}, Q^\top Q = I, \mathbf{a} \in \mathbb{R}^{n_x} \tag{15}$$

where $a, \mathbf{a}, Q$ can be learnable parameters or parametric functions of other network parameters, *e.g.*, $\mathbf{a} = \phi^x(\mathbf{u})$. The matrix $Q$ can also be non orthogonal, provided that the obtained transformation preserves distances or angles.

Devising more complex forms for $\psi$ for arbitrary coordinate embeddings $X, Y$ satisfying the relative distance or angle preserving property, is quite tricky and further assumptions are in general required. To see this, consider a relative distance preserving transformation. Any transformation $X \to Y$, such that $\|\mathbf{x}_i - \mathbf{x}_j\|^2 = \|\mathbf{y}_i - \mathbf{y}_j\|^2$, can be obtained as $\mathbf{x}_i = \gamma_i A_i \mathbf{y}_i + \mathbf{q}_i$, where $\gamma_i$, $A_i$ and $\mathbf{q}_i$, $i \in \mathcal{V}$, are solutions to the system

$$\|\gamma_i A_i \mathbf{y}_i + \mathbf{q}_i - \gamma_j A_j \mathbf{y}_j + \mathbf{q}_j\|^2 = \|\mathbf{y}_i - \mathbf{y}_j\|^2, \quad (i,j) \in \mathcal{E}.$$

with $\gamma_i \in \mathbb{R}$, $A_i^\top A_i = I$, $\mathbf{q}_i \in \mathbb{R}^{n_n}$, $\forall i \in \mathcal{V}$. In this general case, defining a (non trivial) map $\psi$ such that, after its application, one has $\|\mathbf{x}_i^+ - \mathbf{x}_j^+\|^2 = \|\mathbf{y}_i^+ - \mathbf{y}_j^+\|^2 \forall (i,j) \in \mathcal{E}$ is hard without any assumption on, at least, the topology of the graph $\mathcal{G}_X$ (or, equivalently $\mathcal{G}_Y$). A similar reasoning can be applied also to relative angle preserving maps.

If we restrict ourselves to the case in which $\gamma_i = \gamma$, $A_i = A$, $\mathbf{q}_i = \mathbf{q}$, $\forall i \in \mathcal{V}$, this results in $X$ being a Conformal orthogonal transformation of $Y$. In this case, a possible map $\psi$ is defined by

$$\mathbf{x}_i^+ = \mathbf{x}_i + \sum_{j \in \mathcal{N}_i} a_{ji}(\mathbf{x}_j - \mathbf{x}_i). \tag{16}$$

with $a_{ji}$ possibly being a parametric function of node/edge/angle/global attributes, *e.g.*, $a_{ji} = \phi^x\left(\mathbf{e}_{ji}^+, \mathbf{v}_j^+, \mathbf{v}_i^+, \mathbf{u}\right)$. Notice that, under conformal orthogonal transformations (hence also $E(n)$ transformations) $\mathbf{y}_i = \varphi_g(\mathbf{x}) = A\mathbf{x} + b$, $AA^\top = I$, one has that equation 16 is an equivariant map.

In fact,

$$
\begin{aligned}
\mathbf{y}_i^+ &= \mathbf{y}_i + \sum_{j \in \mathcal{N}_i} a_{ji}(\mathbf{y}_j - \mathbf{y}_i) \\
&= \gamma A \mathbf{x}_i + b + \sum_{j \in \mathcal{N}_i} a_{ji}(\gamma A \mathbf{x}_i + b - \gamma A \mathbf{x}_j + b) \\
&= \gamma A \left( \mathbf{x}_i + \sum_{j \in \mathcal{N}_i} a_{ji}(\mathbf{x}_i - \mathbf{x}_j) \right) + b \\
&= \gamma A \mathbf{x}_i^+ + b
\end{aligned}
$$

Thus, when this map is used as $\psi^x$ in the DGN or AGN block, node coordinates can be updated in an equivariant way with respect to conformal orthogonal transformations. Using the same argument, it is easy to show that equation 13 is also equivariant, but equation 14 is not.

Now we show that equation 16 is both relative distance and angle preserving. For the sake of notation, we assume $\gamma = 1$, however the same arguments can be applied when $\gamma \neq 1$.

**Relative distance preservation of equation 16**    To show that equation 16 satisfies the definition of a distance-preserving map it is sufficient to show that, since $\mathbf{x}_i = A\mathbf{y}_i + q$, one has

$$
\begin{aligned}
\mathbf{x}_i^+ &= \mathbf{x}_i + \sum_{j \in \mathcal{N}_i} a_{ij}(\mathbf{x}_j - \mathbf{x}_i) \\
&= A\mathbf{y}_i + \mathbf{q} + \sum_{j \in \mathcal{N}_i} a_{ij}(A\mathbf{x}_j + \mathbf{q} - A\mathbf{x}_i - \mathbf{q}) \\
&= A\mathbf{y}_i + \mathbf{q} + \sum_{j \in \mathcal{N}_i} a_{ij}(A\mathbf{x}_j - A\mathbf{x}_i) \\
&= A \left[ \mathbf{y}_i + \sum_{j \in \mathcal{N}_i} a_{ij}(\mathbf{x}_j - \mathbf{x}_i) \right] + \mathbf{q} \\
&= A\mathbf{y}_i^+ + \mathbf{q}
\end{aligned}
\tag{17}
$$

which implies

$$
\begin{aligned}
\|\mathbf{x}_i^+ - \mathbf{x}_j^+\|^2 &= \|A\mathbf{y}_i^+ + \mathbf{q} - A\mathbf{y}_j^+ - \mathbf{q}\|^2 \\
&= \|A(\mathbf{y}_i^+ - \mathbf{y}_j^+)\|^2 \\
&= (\mathbf{y}_i^+ - \mathbf{y}_j^+)^\top A^\top A (\mathbf{y}_i^+ - \mathbf{y}_j^+) \\
&= \|\mathbf{y}_i^+ - \mathbf{y}_j^+\|^2
\end{aligned}
$$

where in the last line we used the fact that $A^\top A = I$.

**Relative angle preservation of equation 16**    While equation 17 implies that angles are preserved since $\mathbf{x}_i$ is an $E(n)$ transformation of $\mathbf{y}_i$, one can show this explicitly by recalling that the angle between two vectors $\mathbf{x}_j - \mathbf{x}_i$ and $\mathbf{x}_k - \mathbf{x}_i$ can be computed as

$$
\cos\angle(\mathbf{x}_j, \mathbf{x}_i, \mathbf{x}_k) = \frac{(\mathbf{x}_j - \mathbf{x}_i)^\top (\mathbf{x}_k - \mathbf{x}_i)}{\|\mathbf{x}_j - \mathbf{x}_i\| \|\mathbf{x}_k - \mathbf{x}_i\|}
$$

Then, one has

$$
\begin{aligned}
\frac{(\mathbf{x}_j^+ - \mathbf{x}_i^+)^\top (\mathbf{x}_k^+ - \mathbf{x}_i^+)}{\|\mathbf{x}_j^+ - \mathbf{x}_i^+\|\|\mathbf{x}_k^+ - \mathbf{x}_i^+\|} &= \frac{(A\mathbf{y}_j^+ + \mathbf{q} - A\mathbf{y}_i^+ - \mathbf{q})^\top (A\mathbf{y}_k^+ + \mathbf{q} - A\mathbf{y}_i^+ - \mathbf{q})}{\|A\mathbf{y}_j^+ + \mathbf{q} - A\mathbf{y}_i^+ - \mathbf{q}\|\|A\mathbf{y}_k^+ + \mathbf{q} - A\mathbf{y}_i^+ - \mathbf{q}\|} \\
&= \frac{(A\mathbf{y}_j^+ - A\mathbf{y}_i^+)^\top (A\mathbf{y}_k^+ - A\mathbf{y}_i^+)}{\|A\mathbf{y}_j^+ - A\mathbf{y}_i^+\|\|A\mathbf{y}_k^+ - A\mathbf{y}_i^+\|} \\
&= \frac{(\mathbf{y}_j^+ - \mathbf{y}_i^+)^\top A^\top A (\mathbf{y}_k^+ - \mathbf{y}_i^+)}{\sqrt{(\mathbf{y}_j^+ - \mathbf{y}_i^+)^\top A^\top A (\mathbf{y}_j^+ - \mathbf{y}_i^+)}\sqrt{(\mathbf{y}_k^+ - \mathbf{y}_i^+)^\top A^\top A (\mathbf{y}_k^+ - \mathbf{y}_i^+)}} \\
&= \frac{(\mathbf{y}_j^+ - \mathbf{y}_i^+)^\top (\mathbf{y}_k^+ - \mathbf{y}_i^+)}{\|\mathbf{y}_j^+ - \mathbf{y}_i^+\|\|\mathbf{y}_k^+ - \mathbf{y}_i^+\|}
\end{aligned}
$$

where in the last line we used the fact that $A^\top A = I$.

**Local symmetry transformation of equation 16** Suppose we have a local symmetry transformation, $A(\tilde{\mathbf{x}})$ which only acts on a subgraph $\tilde{\mathcal{G}} \in \mathcal{G}$, such that $\mathbf{v} = (\tilde{\mathbf{v}}_1, \tilde{\mathbf{v}}_2, \ldots \tilde{\mathbf{v}}_n, \mathbf{v}_{n+1}, \ldots \mathbf{v}_m)$, and similarly for the edge, coordinate and angle features. The action of $A(\tilde{\mathbf{x}})$ is then

$$
\begin{aligned}
A(\tilde{\mathbf{x}})\mathbf{x} &= (A(\tilde{\mathbf{x}})\tilde{\mathbf{x}}_1, A(\tilde{\mathbf{x}})\tilde{\mathbf{x}}_2, \ldots A(\tilde{\mathbf{x}})\tilde{\mathbf{x}}_n, A(\tilde{\mathbf{x}})\mathbf{x}_{n+1}, \ldots A(\tilde{\mathbf{x}})\mathbf{x}_m) \\
&= (A(\tilde{\mathbf{x}})\tilde{\mathbf{x}}_1, A(\tilde{\mathbf{x}})\tilde{\mathbf{x}}_2, \ldots A(\tilde{\mathbf{x}})\tilde{\mathbf{x}}_n, \mathbf{x}_{n+1}, \ldots \mathbf{x}_m) \ .
\end{aligned}
$$

By defining the 2 subgraphs as containing the coordinate features which are and are not affected by the symmetry transformation, we can therefore write a coordinate update equation 16 for both subgraphs; Eq. equation 16 for the $\mathbf{x}_i$ and, for the $\tilde{\mathbf{x}}_i$,

$$
\tilde{\mathbf{x}}_i^+ = \tilde{\mathbf{x}}_i + \sum_{j \in \mathcal{N}_i} a_{ji}(\tilde{\mathbf{x}}_j - \tilde{\mathbf{x}}_i).
$$

We have shown above that this coordinate update preserves distances and angles for global group transformations; by defining the 2 subgraphs as above, we can promote the local symmetry transformation $A(\tilde{\mathbf{x}})$ to global transformations on subgraphs. Whilst here we have shown this for only 2 subgraphs, one can generalise the argument to any number of local transformations so long as the subgraphs are defined as above. As we have to subdivide the graph into subgraphs, we note that these local transformations cannot be defined arbitrarily as there must be a sense of a neighbourhood of nodes within each subgraph. A local transformation which affects unrelated nodes identically (which is a valid class of local symmetry) is not valid for this reason.

## D    Euclidean group equivariance of the DGN block

To show equivariance to $E(n)$ transformations of the input coordinates for the DGN block, we begin with the edge update in equation 5a. Since $\|\mathbf{x}_i - \mathbf{x}_j\|_2^2$ is invariant under an $E(n)$ transformation, $\mathbf{x} \mapsto Q\mathbf{x} + \mathbf{z}$, for some rotation matrix $Q \in O(n)$ and translation vector $\mathbf{z} \in \mathbb{R}^n$, then

$$
\begin{aligned}
\phi^e\big(\mathbf{e}_{ji}, \mathbf{v}_i, \mathbf{v}_j, \|\mathbf{x}_i - \mathbf{x}_j\|_2^2, \mathbf{u}\big) &\to \phi^e\big(\mathbf{e}_{ji}, \mathbf{v}_i, \mathbf{v}_j, \|Q\mathbf{x}_i + \mathbf{z} - Q\mathbf{x}_j - \mathbf{z}\|_2^2, \mathbf{u}\big) \\
&= \phi^e\big(\mathbf{e}_{ji}, \mathbf{v}_i, \mathbf{v}_j, \|\mathbf{x}_i - \mathbf{x}_j\|_2^2, \mathbf{u}\big) \ .
\end{aligned}
$$

Invariance of the node and global updates in equation 5 follows naturally as they are composed of invariant quantities. The coordinate update in equation 5c can be invariant or equivariant under $E(n)$ depending on the structure of $\psi^x$.

## E    Conformal group invariance of the AGN block

As discussed in Section 2.1, the conformal group consists of transformations that preserve angles between all possible triples of coordinates (see equation 1). To show that the AGN block is invariant

to conformal transformations it is sufficient to note that given a conformal transformation, $\varphi$, one has that the angle update equation 6a is invariant to it since

$$\phi^\alpha(\mathbf{v}_i, \mathbf{v}_j, \mathbf{v}_k, \boldsymbol{\alpha}_{jik}, \angle(\mathbf{x}_j, \mathbf{x}_i, \mathbf{x}_k), \mathbf{u}) \to \phi^\alpha(\mathbf{v}_i, \mathbf{v}_j, \mathbf{v}_k, \boldsymbol{\alpha}_{jik}, \angle(\varphi(\mathbf{x}_j), \varphi(\mathbf{x}_i), \varphi(\mathbf{x}_k)), \mathbf{u})$$
$$= \phi^\alpha(\mathbf{v}_i, \mathbf{v}_j, \mathbf{v}_k, \boldsymbol{\alpha}_{jik}, \angle(\mathbf{x}_j, \mathbf{x}_i, \mathbf{x}_k), \mathbf{u}).$$

Invariance of the other updates is trivially satisfied by construction. The coordinate updates can also be constructed to be equivariant to conformal orthogonal (and hence Euclidean) transformations, with an appropriate choice of $\psi^x$. We stress that equivariance to the conformal group includes by definition equivariance to the Euclidean group as we are only concerned with transformations on $\mathbb{R}^n$. The orthogonal rotations and translations of the Euclidean group are therefore a subset of possible conformal transformations on $\mathbb{R}^n$ as they are angle-conserving transformations. Angular information can be extremely powerful in tasks where classical (or distance-based) GNNs fail, like in graph isomorphism tests where, *e.g.*, a hexagon is not distinguished from two triangles.

## F  ADDITIONAL FORMULATIONS

### F.1  ALTERNATIVE FORMULATIONS FOR THE ANGLE PRESERVING GRAPH NETWORK

A number of variations can be proposed for the angle preserving graph network:

- Edge attributes can be used in angle updates

$$\boldsymbol{\alpha}_{jik}^+ = \phi^\alpha(\mathbf{v}_i, \mathbf{v}_j, \mathbf{v}_k, \mathbf{e}_{ij}, \mathbf{e}_{ik}, \boldsymbol{\alpha}_{jik}, \angle(\mathbf{x}_j, \mathbf{x}_i, \mathbf{x}_k), \mathbf{u}), \qquad \forall(j, i, k) \in \mathcal{A}.$$

- Relative distances can be used in the angle updates

$$\boldsymbol{\alpha}_{jik}^+ = \phi^\alpha(\dots, \|\mathbf{x}_i - \mathbf{x}_j\|_2^2, \|\mathbf{x}_i - \mathbf{x}_k\|_2^2, \angle(\mathbf{x}_j, \mathbf{x}_i, \mathbf{x}_k), \mathbf{u}), \qquad \forall(j, i, k) \in \mathcal{A}.$$

- Angle attributes can be used in edge updates

$$\mathbf{e}_{ij}^+ = \phi^e\big(\mathbf{e}_{ij}, \mathbf{v}_i, \mathbf{v}_j, \rho^{\alpha \to e}(\{\boldsymbol{\alpha}_{ijk}^+\}_{k \in \mathcal{A}_{ij}}), \rho^{\alpha \to e}(\{\boldsymbol{\alpha}_{jik}^+\}_{k \in \mathcal{A}_{ji}}), \mathbf{u}\big), \quad \forall(i, j) \in \mathcal{E}$$

  where $\mathcal{A}_{ij} = \{k \mid (y, z, k) \in \mathcal{A}, y = i, z = j\}$ is the set of angles whose first ray is defined by $(i, j)$.

- Angle embeddings can be ignored and node attributes can be updated with the angles themselves

$$\mathbf{e}_{ji}^+ = \phi^e\big(\mathbf{v}_j, \mathbf{v}_i, \mathbf{e}_{ji}, \mathbf{u}\big), \qquad\qquad\qquad\qquad \forall(j, i) \in \mathcal{E}$$
$$\mathbf{v}_i^+ = \phi^v\big(\mathbf{v}_i, \rho^{e \to v}(\{\mathbf{e}_{ji}^+\}_{j \in \mathcal{N}_i}), \rho^{\alpha \to v}(\{\angle(\mathbf{x}_j, \mathbf{x}_i, \mathbf{x}_k)\}_{(j,k) \in \mathcal{A}_i}), \mathbf{u}\big), \quad \forall i \in \mathcal{V}$$
$$\mathbf{x}_i^+ = \psi^x(i, \mathcal{G}_X), \qquad\qquad\qquad\qquad \forall i \in \mathcal{V}$$
$$\mathbf{u}^+ = \phi^u\big(\rho^{v \to u}(\{\mathbf{v}_i^+\}_{i \in \mathcal{V}}), \rho^{e \to u}(\{\mathbf{e}_{ji}^+\}_{(j,i) \in \mathcal{E}}), \mathbf{u}\big).$$

### F.2  COMBINED ARCHITECTURE

The DGN and AGN architectures can be combined in a single architecture. An example is

$$\boldsymbol{\alpha}_{jik}^+ = \phi^\alpha(\mathbf{v}_i, \mathbf{v}_j, \mathbf{v}_k, \boldsymbol{\alpha}_{jik}, \angle(\mathbf{x}_j, \mathbf{x}_i, \mathbf{x}_k), \mathbf{u}), \qquad\qquad \forall(j, i, k) \in \mathcal{A}$$
$$\mathbf{e}_{ji}^+ = \phi^e\big(\mathbf{v}_j, \mathbf{v}_i, \mathbf{e}_{ji}, \|\mathbf{x}_i - \mathbf{x}_j\|_2^2, \mathbf{u}\big), \qquad\qquad \forall(j, i) \in \mathcal{E}$$
$$\mathbf{v}_i^+ = \phi^v\big(\mathbf{v}_i, \rho^{e \to v}(\{\mathbf{e}_{ji}^+\}_{j \in \mathcal{N}_i}), \rho^{\alpha \to v}(\{\boldsymbol{\alpha}_{jik}^+\}_{(j,k) \in \mathcal{A}_i}), \mathbf{u}\big), \qquad \forall i \in \mathcal{V}$$
$$\mathbf{x}_i^+ = \psi^x(i, \mathcal{G}_X), \qquad\qquad\qquad\qquad \forall i \in \mathcal{V}$$
$$\mathbf{u}^+ = \phi^u\big(\rho^{v \to u}(\{\mathbf{v}_i^+\}_{i \in \mathcal{V}}), \rho^{e \to u}(\{\mathbf{e}_{ji}^+\}_{(j,i) \in \mathcal{E}}), \mathbf{u}\big)$$

where the global update can contain aggregated information about angles and distances and also the other updates can be generalised as shown above. This architecture is by construction equivariant to transformations in the coordinate embeddings for which both relative distances and angles are preserved.

# G  OBTAINING OTHER ARCHITECTURES AS SPECIAL INSTANCES

In this appendix we will represent some of the architectures discussed in the main text explicitly as instances of our architecture.

## G.1  DIMENET (KLICPERA ET AL., 2020)

Dimenet can be obtained from the AGN by considering edge and distance information in the angle update and using sum aggregation functions as

$$
\begin{aligned}
\boldsymbol{\alpha}_{ijk}^{+} &= \phi^{\alpha}(\mathbf{e}_{ji}, \|\mathbf{x}_i - \mathbf{x}_j\|_2^2, \angle(\mathbf{x}_i, \mathbf{x}_j, \mathbf{x}_k)), \ \ \forall(j,i,k) \in \mathcal{A} \\
\mathbf{e}_{ji}^{+} &= \phi^{e}\big(\mathbf{e}_{ji}, \sum_{i \in \mathcal{N}_j} \boldsymbol{\alpha}_{ijk}^{+}\big), && \forall(j,i) \in \mathcal{E} \\
\mathbf{x}_i^{+} &= \mathbf{x}_i, && \forall i \in \mathcal{V} \\
\mathbf{v}_i^{+} &= \phi^{v}\big(\mathbf{v}_i, \sum_{j \in \mathcal{N}_i} \mathbf{e}_{ji}^{+}\big), && \forall i \in \mathcal{V}
\end{aligned}
$$

where $\|\mathbf{x}_i - \mathbf{x}_j\|_2^2$ is defined represented within a set of orthogonal basis functions $e_{RBF}$ and the angles $\angle(\mathbf{x}_i, \mathbf{x}_j, \mathbf{x}_k)$ within a basis defined as $\alpha_{SBF}$.

## G.2  EGNN (SATORRAS ET AL., 2021)

The EGNN network is obtained from the DGN by selecting a specific form for the coordinate update function $\psi^x$, using the sum aggregation function as $\phi^{e \to v}$, and not propagating updated edges, *i.e.*,

$$
\begin{aligned}
\mathbf{e}_{ji}^{+} &= \phi^{e}\big(\mathbf{v}_j, \mathbf{v}_i, \mathbf{e}_{ji}^{in}, \|\mathbf{x}_i - \mathbf{x}_j\|_2^2\big), && \forall(j,i) \in \mathcal{E}, \\
\mathbf{x}_i^{+} &= \mathbf{x}_i + \sum_{j \neq i}(\mathbf{x}_i - \mathbf{x}_j)\phi^{x}(\mathbf{e}_{ji}^{+}), && \forall i \in \mathcal{V}, \\
\mathbf{v}_i^{+} &= \phi^{v}\big(\mathbf{v}_i, \sum_{j \in \mathcal{N}_i} \mathbf{e}_{ji}^{+}\big) && ,\forall i \in \mathcal{V},
\end{aligned}
$$

where $\mathbf{e}_{ji}^{in}$ are the edge attributes of the input data (implying that $\mathbf{e}_{ji}^{+}$ is not propagated to any subsequent layer, as in message passing networks).

## G.3  ISOGNN (HORIE ET AL., 2021)

The IsoGNN architecture is defined for tensors of rank-$n$; to compare with the other architectures presented here, we show below the architecture for rank-1 tensors.

$$
\begin{aligned}
\mathbf{e}_{ji}^{+} &= \left( \sum_{k,l \in \mathcal{V}, k \neq l} \mathbf{T}_{ijkl}(\mathbf{x}_k - \mathbf{x}_l) \right) \mathbf{v}_j, \ \ \forall(j,i) \in \mathcal{E} \\
\mathbf{x}_i^{+} &= \mathbf{x}_i, && \forall i \in \mathcal{V} \\
\mathbf{v}_i^{+} &= \phi^{v}\big(\mathbf{v}_i, \sum_{j \in \mathcal{N}_i} \mathbf{e}_{ji}^{+}\big), && \forall i \in \mathcal{V}
\end{aligned}
$$

where $\mathbf{T}_{ijkl}$ is an untrainable 2-dimensional matrix which is translation and rotation invariant, and determined offline from the data for each class of problem.

## G.4  OTHER NETWORKS

Standard graph networks can be obtained from ours by skipping some updates or not considering equivariant information. Also other variants, including SchNet (Schütt et al., 2017) or TFN (Thomas et al., 2018), can be cast as message passing architectures as shown in (Satorras et al., 2021).

## H  ADDITIONAL EXPERIMENTS AND IMPLEMENTATION DETAILS

### H.1  COMMON IMPLEMENTATION DETAILS

The update functions of the networks are all implemented as MLPs. After the graph layers, the produced node embeddings are passed through another MLP, a global pooling layer and a final MLP with output dimension equal to the number of classes for graph classification tasks. For edge classification tasks (TSP), the architecture after the graph layers is a single MLP taking as input source and target nodes and predicting the class of each edge. Each network is trained starting from 10 different initial conditions. The results in the tables contains the mean and standard deviation resulting from the 10 initialisations.

### H.2  POLYTOPES CLASSIFICATION

#### H.2.1  SPECIFIC IMPLEMENTATION DETAILS

All the MLPs have one hidden layer containing 64 neurons and swish activation function. We used 2 graph layers for AGN and DimeNet and 3 for the DGN, GN and EGNN with embeddings in the hidden layers having dimension 32. Aggregation functions and the pooling layer implements mean or sum operations (and are specified in the results' tables) for our architectures. As for DimeNet, we use distances and angles directly instead of their RBF embeddings since we do not consider atomistic quantities. Adam (Kingma & Ba, 2015) is used to train the all the models with a learning rate $\alpha = 0.001$ and no regularisation for 1000 epochs. The batch-size is equal to the number of training samples (so $5, 6, 3$ respectively, for $n = 3, 4, 5$).

#### H.2.2  ADDITIONAL EXPERIMENTS

In the main paper we reported results when the identity function equation 12 was used to perform the coordinate update together with a sum aggregation function. In Tables 4, 5 and 6 we report also the results obtained when using equation 16 for the coordinate updates and possibly mean aggregation function. As can be seen, using the mean aggregation function usually causes a drop in performance. In particular, the SDGN with mean aggregation function is unable to correctly classify the polytopes even at training time. This is because, after rescaling the coordinates, all edges have the same length and, employing a mean aggregation results in always the same output for all polytopes. This is partially alleviated by using equation 16 as the node coordinate update function, which allows one to differently remap coordinates of different polytopes. The other results are qualitatively similar.

| | | | | test accuracy | | | | |
| block | $\rho$ | $\psi^x$ | train acc | Orthogonal | Orthogonal + dilation | Non-orthogonal ($\mu = 0.5$) | Non-orthogonal ($\mu = 1.5$) | Non-orthogonal ($\mu = 3.0$) |
|---|---|---|---|---|---|---|---|---|
| AGN | mean | 12 | 1 | **1** | **1** | **1** | **1** | **$0.96 \pm 0.04$** |
| AGN | mean | 16 | 1 | **1** | **1** | **1** | **1** | **$0.97 \pm 0.03$** |
| AGN | sum | 12 | 1 | **1** | **1** | **1** | **1** | **1** |
| AGN | sum | 16 | 1 | **1** | **1** | **1** | **1** | **1** |
| SDGN | mean | 12 | 0.2 | 0.2 | 0.2 | 0.2 | 0.2 | 0.2 |
| SDGN | mean | 16 | 0.8 | 0.8 | 0.8 | $0.65 \pm 0.05$ | $0.40 \pm 0.08$ | $0.36 \pm 0.06$ |
| SDGN | sum | 12 | 1 | **1** | **1** | **1** | **$0.93 \pm 0.07$** | $0.83 \pm 0.15$ |
| SDGN | sum | 16 | 1 | **1** | **1** | **$0.89 \pm 0.16$** | $0.62 \pm 0.24$ | $0.60 \pm 0.21$ |
| DGN | mean | 12 | 1 | 0.83 | $0.22 \pm 0.02$ | $0.22 \pm 0.03$ | $0.20 \pm 0.01$ | $0.22 \pm 0.04$ |
| DGN | mean | 16 | 1 | **1** | $0.29 \pm 0.02$ | $0.25 \pm 0.02$ | $0.24 \pm 0.02$ | $0.20 \pm 0.03$ |
| DGN | sum | 12 | 1 | **1** | $0.45 \pm 0.05$ | $0.44 \pm 0.12$ | $0.41 \pm 0.07$ | $0.37 \pm 0.04$ |
| DGN | sum | 16 | 1 | **1** | $0.43 \pm 0.05$ | $0.41 \pm 0.04$ | $0.43 \pm 0.08$ | $0.40 \pm 0.03$ |
| GN | mean | — | 1 | $0.25 \pm 0.04$ | $0.20 \pm 0.05$ | $0.21 \pm 0.03$ | $0.20 \pm 0.02$ | $0.22 \pm 0.04$ |
| GN | sum | — | 1 | $0.44 \pm 0.15$ | $0.46 \pm 0.14$ | $0.47 \pm 0.15$ | $0.44 \pm 0.17$ | $0.43 \pm 0.15$ |

Table 4: Polytopes classification: training and test accuracy (mean $\pm$ standard deviation over 10 runs) for $n = 3$, for different transformations in the test set.

**Adding not all-identical node features**  The experiments run so far used datasets containing only information about node coordinates and the presence of edges. If we add (not-all-identical) node features, then also SDGN with mean aggregation function is able to correctly classify all the polytopes. This happens thanks to the additional features breaking a symmetry making the graphs of the simplex and the orthoplex look identical to the SDGN layer.

| block | $\rho$ | $\psi^x$ | train acc | test accuracy | | | | |
|---|---|---|---|---|---|---|---|---|
| | | | | Orthogonal | Orthogonal + dilation | Non-orthogonal ($\mu = 0.5$) | Non-orthogonal ($\mu = 1.5$) | Non-orthogonal ($\mu = 3.0$) |
| AGN | mean | 12 | 1 | **1** | **1** | **1** | **1** | **0.98 ± 0.04** |
| AGN | mean | 16 | 1 | **1** | **1** | **1** | **1** | **0.97 ± 0.04** |
| AGN | sum | 12 | 1 | **1** | **1** | **1** | **1** | **1** |
| AGN | sum | 16 | 1 | **1** | **1** | **1** | **1** | **1** |
| SDGN | mean | 12 | 0.17 | 0.17 | 0.17 | 0.17 | 0.17 | 0.17 |
| SDGN | mean | 16 | 1 | **1** | **1** | 0.57 ± 0.13 | 0.35 ± 0.10 | 0.27 ± 0.10 |
| SDGN | sum | 12 | 1 | **1** | **1** | 0.96 ± 0.03 | 0.83 ± 0.13 | 0.77 ± 0.17 |
| SDGN | sum | 16 | 1 | **1** | **1** | 0.92 ± 0.06 | 0.78 ± 0.14 | 0.65 ± 0.20 |
| DGN | mean | 12 | 0.83 | 0.83 | 0.31 ± 0.02 | 0.40 ± 0.03 | 0.32 ± 0.01 | 0.27 ± 0.04 |
| DGN | mean | 16 | 1 | **1** | 0.34 ± 0.02 | 0.40 ± 0.04 | 0.38 ± 0.03 | 0.30 ± 0.03 |
| DGN | sum | 12 | 1 | **1** | 0.61 ± 0.05 | 0.60 ± 0.08 | 0.61 ± 0.06 | 0.59 ± 0.08 |
| DGN | sum | 16 | 1 | **1** | 0.60 ± 0.04 | 0.61 ± 0.03 | 0.57 ± 0.08 | 0.55 ± 0.07 |
| GN | mean | – | 1 | 0.18 ± 0.04 | 0.19 ± 0.05 | 0.21 ± 0.03 | 0.24 ± 0.02 | 0.22 ± 0.04 |
| GN | sum | – | 1 | 0.53 ± 0.04 | 0.51 ± 0.03 | 0.48 ± 0.12 | 0.49 ± 0.06 | 0.51 ± 0.04 |

Table 5: Polytopes classification: training and test accuracy (mean ± standard deviation over 10 runs) for $n = 4$, for different transformations in the test set.

| block | $\rho$ | $\psi^x$ | train acc | test accuracy | | | | |
|---|---|---|---|---|---|---|---|---|
| | | | | Orthogonal | Orthogonal + dilation | Non-orthogonal ($\mu = 0.5$) | Non-orthogonal ($\mu = 1.5$) | Non-orthogonal ($\mu = 3.0$) |
| AGN | mean | 12 | 1 | **1** | **1** | **0.99 ± 0.03** | 0.86 ± 0.11 | 0.77 ± 0.13 |
| AGN | mean | 16 | 1 | **1** | **1** | **0.99 ± 0.03** | 0.84 ± 0.11 | 0.75 ± 0.11 |
| AGN | sum | 12 | 1 | **1** | **1** | **1** | **1** | **1** |
| AGN | sum | 16 | 1 | **1** | **1** | **1** | **1** | **1** |
| SDGN | mean | 12 | 0.33 | 0.33 | 0.33 | 0.33 | 0.33 | 0.33 |
| SDGN | mean | 16 | 1 | **1** | **1** | 0.60 ± 0.25 | 0.48 ± 0.15 | 0.37 ± 0.10 |
| SDGN | sum | 12 | 1 | **1** | **1** | **0.99 ± 0.01** | **0.98 ± 0.02** | **0.98 ± 0.02** |
| SDGN | sum | 16 | 1 | **1** | **1** | 0.83 ± 0.16 | 0.75 ± 0.14 | 0.66 ± 0.18 |
| DGN | mean | 12 | 1 | **1** | 0.49 ± 0.01 | 0.33 ± 0.00 | 0.49 ± 0.02 | 0.37 ± 0.01 |
| DGN | mean | 16 | 1 | **1** | 0.51 ± 0.03 | 0.40 ± 0.04 | 0.41 ± 0.06 | 0.39 ± 0.05 |
| DGN | sum | 12 | 1 | **1** | 0.46 ± 0.05 | 0.44 ± 0.05 | 0.39 ± 0.08 | 0.41 ± 0.10 |
| DGN | sum | 16 | 1 | **1** | 0.57 ± 0.10 | 0.45 ± 0.12 | 0.54 ± 0.13 | 0.49 ± 0.10 |
| GN | mean | – | 1 | 0.38 ± 0.05 | 0.42 ± 0.08 | 0.34 ± 0.04 | 0.38 ± 0.08 | 0.43 ± 0.06 |
| GN | sum | – | 1 | 0.64 ± 0.07 | 0.55 ± 0.04 | 0.60 ± 0.13 | 0.61 ± 0.16 | 0.58 ± 0.08 |

Table 6: Polytopes classification: training and test accuracy (mean ± standard deviation over 10 runs) for $n = 5$, for different transformations in the test set.

**Data efficiency**    To emphasise the advantage of having a network that is able to exploit symmetries in the dataset in terms of data efficiency, we study how many samples in the training set are necessary for a standard GNN to reach reasonable generalisation performance. For the set of transformations we considered in the previous sections, we augmented the training set with $\{2, 3, \ldots, 100\}$ randomly transformed (as in the respective test set) copies of each polytope. We trained a standard GNN on these augmented datasets and observed the resulting test accuracy after 1000 epochs. Results are reported in Figure 4 for each set of transformations in terms of mean and standard deviation over 10 random initialisations. It can be seen that 20 samples per polytope may be sufficient when only orthogonal transformations are considered. Adding also dilations and non-orthogonal transformations further increases the number of data points that are required. This shows that while data augmentation can be successfully exploited, it is provably sub-optimal in terms of sample complexity (Mei et al., 2021) and architectures with built-in equivariance properties represent a more efficient strategy to consider.

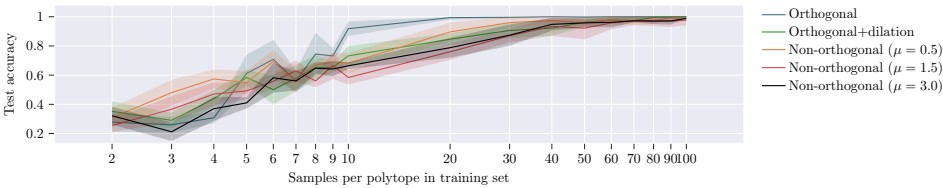

Figure 4: Test accuracy vs samples per polytope in the training set for a standard GNN ($n = 3$).

## H.3   MNIST, CIFAR10, TSP

MNIST and CIFAR10 are classical image classification datasets converted into graphs using super-pixels and assigning the super-pixel coordinates as node coordinates and the intensity as node fea-

tures. TSP (based on the Travelling Salesman Problem) tests link prediction on 2D Euclidean graphs to identify edges belonging to the optimal TSP solution given by the Concorde solver.

For this datasets we use the same splits as specified in (Dwivedi et al., 2020).

### H.3.1 SPECIFIC IMPLEMENTATION DETAILS

The MLPs have one hidden layer containing 64 neurons, swish activation function and dropout layers with dropout rate 0.01. We used 2 graph layers for AGN and 3 for both the DGN and GN with embeddings in the hidden layers having dimension 64. Aggregation functions and the pooling layer implements sum operations. The models are trained with Adam with learning rate $\alpha = 0.001$ and no regularisation for 100 epochs. Batch-size 128 is used for MNIST and CIFAR10, and 8 for TSP.

### H.4 MOLECULAR PROPERTY PREDICTION - QM9

The QM9 dataset (Wu et al., 2017) is comprised of small molecules (hydrogen, carbon, nitrogen, oxygen, flourine) with the target properties being 12 chemical properties. As the target properties are equivariant to Euclidean transformations of the atoms' coordinates, and also to the order in which atoms are processed, QM9 is an excellent benchmarking dataset for a GNN, especially if $E(n)$ invariant. Indeed, state of the art results have been achieved on this dataset by EGNN, Dimenet, SE3-Transformer (Satorras et al., 2021; Klicpera et al., 2020; Fuchs et al., 2020). Here we show that, in addition to leading to better accuracy, employing equivariant networks give a significant improvement in convergence speed.

Figure 5 show the evolution of the mean squared error on the test set on all the target properties for both architectures. We see that both the AGN and DGN outperform the standard GN in two aspects; the model trains more rapidly, and also reaches a lower loss.

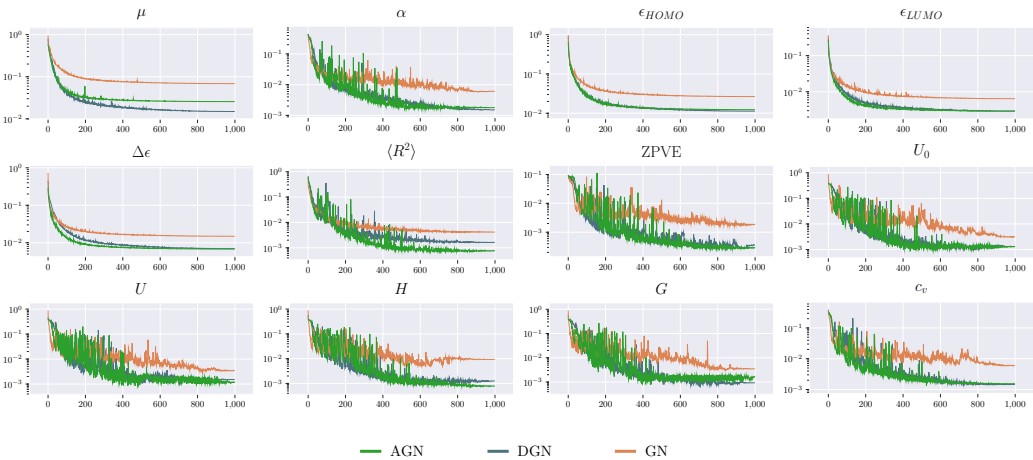

Figure 5: QM9: test MSE loss on the 12 target properties.

### H.4.1 SPECIFIC IMPLEMENTATION DETAILS

The QM9 dataset is composed of roughly 134k molecules: we used 100k for training and the remaining for testing. The AGN and DGN networks receive the embedding of the atomic properties as initial node features and the coordinates of each atom as coordinate features, while in the standard GNN they are stacked and provided as node features. Edge embeddings representing bond types are also provided.

The networks consist of 4 graph layers (3 for the AGN) with each MLP having 2 hidden layers of 128 nodes, swish activation function and dropout rate of 0.01. The node embeddings of the last graph layer are passed through an MLP, a global mean pooling layer and a last MLP to map them to the target chemical property. The last two MLPs each have a single hidden layer of 128 nodes, swish

activation and dropout rate of 0.01. The target chemical properties are all standardised by subtracting the mean and dividing by the standard deviation for each target. The networks are trained for 1000 epochs using ADAM with a learning rate of $0.0005$ and mean squared error loss.

