# OpenReview forum: "Symmetry-driven graph neural networks"
_ICLR.cc/2022/Conference — ICLR 2022 Submitted_

### Official Review · Reviewer_W7xU · 2021-10-30

**Correctness:** 4
**Technical Novelty And Significance:** 3
**Empirical Novelty And Significance:** 3
**Recommendation:** 6
**Confidence:** 4

**Main Review:**

This paper proposes DGN and AGN that can update graph representation equivariantly w.r.t. any transformation preserving distance and angles, respectively. The authors conduct experiments on synthetic data to demonstrate the effectiveness of considering symmetries.

#####Pros#####

1. It is intuitively sound and necessary to consider the symmetries of data. The motivation of this work is clearly explained and convincing.


2. The proposed DGN and AGN, shown in Figure 2 (b) and (c), are technically sound and general enough. They are developed based on the general framework GN (Figure 2 (a)). Most existing methods, such as SchNet and DimeNet included in Section 5, can be viewed as special cases of the proposed frameworks. Hence, the proposed DGN and AGN are unified well.


3. The experiments on synthetic data are well-designed, which can show the effectiveness of exploiting symmetries. This can support the methodology strongly.


#####Cons#####

1. The main concern is that there are no empirical results to demonstrate that DGN or AGN is effective on real-world tasks, instead of specifically designed synthetic tasks. It is strongly desired that the proposed method can be evaluated on real datasets.


**Summary Of The Paper:**

This paper proposes two graph networks that are equivariant to distance and angle preserving transformations in graph coordinates. It moves one step forward based on GN (Graph Network) by decoupling node coordinates from other node attributes. The experiments on synthetic datasets shows the capabilities of the proposed symmetry-driven graph networks.

**Summary Of The Review:**

Overall, I think the current version of this paper is slightly above the acceptance threshold since its motivation and proposed methods are technically sound and convincing.

---

> ### Author Response · Authors · 2021-11-09
> **Rebuttal to Reviewer W7xU**
>
> We thank the reviewer for their positive review and valuable comments and suggestions. Detailed answers to the points raised by the reviewer are provided below.
>
> > Pros
> > 1. It is intuitively sound and necessary to consider the symmetries of data. The motivation of this work is clearly explained and convincing.
> > 2. The proposed DGN and AGN, shown in Figure 2 (b) and (c), are technically sound and general enough. They are developed based on the general framework GN (Figure 2 (a)). Most existing methods, such as SchNet and DimeNet included in Section 5, can be viewed as special cases of the proposed frameworks. Hence, the proposed DGN and AGN are unified well.
> > 3. The experiments on synthetic data are well-designed, which can show the effectiveness of exploiting symmetries. This can support the methodology strongly.
>
> Thank you for these positive comments and for appreciating the synthetic experiments. Also, we have indeed aimed to show that our framework can be viewed as a unifying architecture of many existing methods currently in the literature.
>
> > Cons
> > 1. The main concern is that there are no empirical results to demonstrate that DGN or AGN is effective on real-world tasks, instead of specifically designed synthetic tasks. It is strongly desired that the proposed method can be evaluated on real datasets.
>
> We performed experiments on the following real-world datasets: TSP, MNIST, CIFAR10 in the main paper and QM9 in the appendix, section H. We used the first three to show specific use-cases where using an equivariant architecture can be counterproductive – for example, MNIST and CIFAR10 are datasets with a lot of symmetry, but the embedding of the images as graphs (Dwivedi et al., 2020) causes these symmetries to be lost, as we discuss in Section 6.2. Regarding QM9, we used it to show how being invariant or equivariant allows architectures such as the ones we presented to achieve a gain in convergence speed with respect to a standard graph network.
> We want to point out that the application of the proposed architectures to other datasets is the subject of ongoing work.

---

> > ### Comment · Reviewer_W7xU · 2021-11-11
> > **Thanks for the response**
> >
> > Thank you for answering the question. I am currently having no other questions and will discuss with other reviewers and ACs.

---

### Official Review · Reviewer_yqP4 · 2021-11-03

**Correctness:** 4
**Technical Novelty And Significance:** 3
**Empirical Novelty And Significance:** 3
**Recommendation:** 6
**Confidence:** 4

**Main Review:**

Initial Recommendation: Marginal Accept

I have specified the reasons for my recommendation below:

Strengths:
1. The paper is clearly structured and well written, and I found it easy to read.
2. Analogous to E(N) GNN - the model is simple and computationally efficient (along with quantitative results which show computational complexity) - without the use of special functions like Bessel functions, etc to capture equivariance to the desired transformation actions.
3. Strong experimental results on the synthetic polytope classification datasets when the symmetries are present - and also provide the reader examples of when the model should not be used.

Weakness:
1. The authors discuss the ability to capture molecular conformers as one of the main selling points of the network being equivariant to the conformal group but do not show results on the QM9/ other macromolecular datasets in comparison to E(N) GNN, but without using a fully connected graph (Apart from rate of convergence in the appendix). For instance, in molecules, only when covalent/ hydrogen bonds are present does the distance between atoms need to be preserved, right?.
2. The authors do not present other use cases (or motivations of this work) of the symmetries to the conformal group or distance preserving transformations of an object - for example results on point cloud datasets (where the objects are not rigid bodies) would do well for this work.


Minor:
1. Introduce $a_{ji}$ earlier rather than in page 4 as it is used in eq 3 (in pg 3)

**Summary Of The Paper:**

The authors propose distance preserving Graph Network and Angle Preserving Graph Networks which are equivariant to node permutation as well distance preserving transformations/ angle preserving transformations of the coordinates associated with the nodes. The authors generalize the E(N) GNN work to include group symmetries to the conformal group and to the case when not all nodes in the graph are connected. Experiments on the  polytopes classification dataset show that the model is effective when symmetries are present in the data, and ineffective when they are not (MNIST, CIFAR and TSP).

**Summary Of The Review:**

The paper is well written and the proposed solution is simple and computationally efficient. The claims are sound and the empirical evaluation is partly adequate but partly needs work.

---

> ### Author Response · Authors · 2021-11-09
> **Rebuttal to Reviewer yqP4**
>
> We thank the reviewer for the positive review and the provided comments and suggestions. We address their concerns below.
>
> > Strengths:
> > 1. The paper is clearly structured and well written, and I found it easy to read.
> > 2. Analogous to E(N) GNN - the model is simple and computationally efficient (along with quantitative results which show computational complexity) - without the use of special functions like Bessel functions, etc to capture equivariance to the desired transformation actions.
> > 3. Strong experimental results on the synthetic polytope classification datasets when the symmetries are present - and provide the reader examples of when the model should not be used.
>
> Thank you for the positive comments and for recognising the importance of computational efficiency with respect to other methods where expensive transformations (like Bessel functions) are applied on node features. We also totally agree that it’s always important to showcase model strengths along with its limitations.
>
> > Weaknesses
> > 1. The authors discuss the ability to capture molecular conformers as one of the main selling points of the network being equivariant to the conformal group but do not show results on the QM9/ other macromolecular datasets in comparison to E(N) GNN, but without using a fully connected graph (Apart from rate of convergence in the appendix). For instance, in molecules, only when covalent/ hydrogen bonds are present does the distance between atoms need to be preserved, right?
>
> We thank the reviewer for this comment, and they make a very interesting point. The reason we use the molecular graph structure and preserve the distance between atoms is because the molecules presented in the QM9 dataset are all in their equilibrium positions. We therefore can assume translational and rotational invariance in the Coulomb matrix, and do not need to take into consideration vibrational modes, which will affect atomic distances. For other macromolecular datasets, this condition may not hold, however QM9 is the ‘standard’ dataset against which equivariant architectures are usually benchmarked.
>
> > 2. The authors do not present other use cases (or motivations of this work) of the symmetries to the conformal group or distance preserving transformations of an object - for example results on point cloud datasets (where the objects are not rigid bodies) would do well for this work.
>
> Due to space limitations, we presented the polytopes classification example and the one on MNIST/CIFAR10/TSP in the main paper (and QM9 in the appendix). This choice is motivated by the fact that we believe that it's important to show both the strengths and limitations of the proposed architectures on some easy-to-visualize datasets in a pedagogical paper.
>
> On the one hand, the polytopes example provides a very clear comparison between differing architectures and enables an intuitive understanding of the importance of considering symmetries on such a dataset. On the other hand, we show how performance may deteriorate when architectures like the ones we presented are applied to datasets in which the expected symmetries are not present to be as transparent as possible with respect to the limitations of such architectures. In addition, we must say that the application of the proposed architectures to other datasets is the subject of ongoing work. If the reviewer has any specific dataset in mind, we'll make sure to consider it as well.
>
> > Minor
> > 1. Introduce $a_{ij}$ earlier rather than in page 4 as it is used in eq 3 (in pg 3)
>
> Thank you for pointing that out. We defined $a_{ji}$ after equation (3) in the revised version of the manuscript.

---

> > ### Comment · Reviewer_yqP4 · 2021-11-11
> > **Reply to the authors**
> >
> > Thank you very much for the reply.
> >
> > With regard to QM9 and other macromolecular datasets -
> >
> > i) Could you please add the Table (in Appendix) for QM9 with the 12/13 different tasks (I am unable to locate it right now) and provide comparisons with E(N) GNN in particular. Specifically, rather than assuming that the graph is fully connected or using a radii of 5 angstrom to construct the 1 hop node neighborhood, I would suggest varying the radii in {2,3,4,5} and showcasing the effect of using the DGN/ AGN. I agree with your point that the QM9 atom positions are equilibrium states for the molecules (and the results might not always be favorable) - but these results would further emphasize your claims.
> >
> > ii)  For the macromolecular datasets - I would suggest adding results on the datasets from Atom3D (https://www.atom3d.ai/) and provide comparisons with GVP-GNN [1][2]. This would further validate your claims about 'learning representations which capture the conformers of macromolecules'.
> >
> > References:
> > [1] Jing, Bowen, et al. "Learning from protein structure with geometric vector perceptrons." arXiv preprint arXiv:2009.01411 (2020).
> > [2] Jing, Bowen, et al. "Equivariant Graph Neural Networks for 3D Macromolecular Structure." arXiv preprint arXiv:2106.03843 (2021).

---

> > > ### Author Response · Authors · 2021-11-11
> > > **Reply to Reviewer yqP4**
> > >
> > > Thank you for the additional comments, which clearly point in the direction of our ongoing research. We address them below.
> > >
> > > > i) Could you please add the Table (in Appendix) for QM9 with the 12/13 different tasks (I am unable to locate it right now) and provide comparisons with E(N) GNN in particular. Specifically, rather than assuming that the graph is fully connected or using a radii of 5 angstrom to construct the 1 hop node neighborhood, I would suggest varying the radii in {2,3,4,5} and showcasing the effect of using the DGN/ AGN. I agree with your point that the QM9 atom positions are equilibrium states for the molecules (and the results might not always be favorable) - but these results would further emphasize your claims.
> > >
> > > Additional experiments on QM9 and other molecular datasets (like MD17) are currently a focus of investigation, as we agree a thorough benchmarking exercise against other architectures is warranted. Regarding QM9, due to the requirement to train and fine-tune 12 separate architectures (one per target property), this is a computationally very intensive task, and we do not envisage the work being completed in time for final submissions to ICLR. However, it will be the subject of a follow-up application paper with in-depth analysis and comparisons of the AGN and DGN architectures applied to QM9 and other datasets.
> > >
> > > > ii) For the macromolecular datasets - I would suggest adding results on the datasets from Atom3D (https://www.atom3d.ai/) and provide comparisons with GVP-GNN [1][2]. This would further validate your claims about 'learning representations which capture the conformers of macromolecules'.
> > >
> > > We thank the reviewer for this suggestion. Whilst we are considering some of the datasets in Atom3D for a follow-up application paper (where we'll also compare against GVP-GNN), we argue this comparison is beyond the scope of this paper. In fact, we aimed to introduce our novel architectures in a pedagogical way, with experiments serving to highlight positive and negative use-cases of such equivariant architectures. Whilst our architectures are naturally applicable to molecular datasets, we feel that focusing on multiple macromolecular datasets would falsely narrow the apparent application space of our architectures.

---

### Official Review · Reviewer_ECjd · 2021-11-03

**Correctness:** 3
**Technical Novelty And Significance:** 2
**Empirical Novelty And Significance:** 2
**Recommendation:** 5
**Confidence:** 4

**Main Review:**

Strengths:

1.	This paper is well written. I enjoy the reading. The authors have introduced the necessary conceptions in a comfortable way. It is easy to follow the whole idea and the methodology.

2.	It is good that the proposed DGN and AGN can generalize the current methods including EGNN and DIMENET, by implementing different choices of certain components. The validity of preserving the distance or angles is clearly justified (mainly in appendix).

3.	The experiments on polytopes classification, though seems over-simplified, are able to demonstrate the superiority of the proposed AGN over other methods (GN, SE3-Trans, DimeNET, EGNN) particularly when non-orthogonal symmetry is considered.

Weaknesses:

1.	The novelty is somehow limited. It is indeed that Equations (5-6) have extended the design of GN (Eq. (4)) by involving equivariance w.r.t. node coordinates and invariance w.r.t. other embedding features. But the main step contributed to this property lies in the update of the node coordinates, namely, Eq. (5c) and Eq. (6d), which is previously developed by EGNN (Satorras et al., 2021) and other related works. This can be checked by comparing Eq. (3) used in this paper with Eq. (4) in the EGNN paper. Even the authors have additionally proved that Eq. (3) is equivariant to Conformal orthogonal transformations which is not discussed in EGNN, such equivariance is trivial to derive. Specifically, for DGN, it is very similar to EGNN, unless that DGN is built upon GN (with global vector u) while EGNN is upon MPNN.

2.	The proposed AGN and DGN show merits only on the simulated dataset. For the real datasets (MNIST, CIFAR10, and TSP), the performance of AGN and DGN are even worse than GN. The authors attribute this detriment to the lack of symmetry in the data. This seems somewhat questionable. Since the images for example in MNIST are actually rotation/translation equivariant, DGN is supposed to perform better than GN, given that DGN is a rotation/translation equivariant version of GN by the design in Eq.(5). Previous studies such as group CNN have also demonstrated further taking the rotation equivariance into account delivers desirable enhancement. The authors are suggested to provide more explanations on this issue.

3.	In terms of evaluations on polytopes classification, all methods achieve accuracy 1 for training. Does this mean that the simulated dataset is too simple? What is the number of the training/testing samples? Is it too small?

4.	There remain several confusing presentations in the current version:

4.1	In the first paragraph, the authors state that graph neural networks, are invariant to the symmetry groups. The “symmetry group” here is too general and includes other cases besides the permutation group.

4.2	It seems AGN is only equivariant with regard to the Conformal orthogonal group (a subgroup of Conformal transformations) according to the derivations in Appendix C. So why has the claim that AGN is equivariant to the Conformal group?

4.3	The authors are suggested to provide formal definitions of these groups: E(n), CO(Rn;Q) and Conf(Rn;0) groups.


**Summary Of The Paper:**

This paper proposes two kinds of equivariant GNNs, including the distance preserving graph network (DGN) that is equivariant to the Euclidean transformations and angle preserving graph network (AGN) that is equivariant to the Conformal group. The proposed frameworks generalize several typical previous architectures. The advantage is experimentally evaluated on a synthetic dataset composed of n-dimensional geometric objects.

**Summary Of The Review:**

Overall, there are certain merits for the proposed architectures, but both the originality and the experimental significance are insufficient. I initially suggest weak rejection.

---

> ### Author Response · Authors · 2021-11-09
> **Rebuttal to Reviewer ECjd**
>
> We thank the reviewer for their valuable comments and suggestions. Detailed answers to the points raised by the reviewer are provided below.
>
> > Strengths:
> >
> > [...]
>
> Thank you for the positive comments. We chose to place the mathematical justifications in the appendix to make the paper readable for those not interested in the derivations. Moreover, whilst we agree the polytopes classification experiment could be considered a simple application of the architectures, it allows us to clearly demonstrate individual aspects of the invariance under $E(n)$ and $Conf(R^n; 0)$.
>
> > Weaknesses:
> >
> > 1. The novelty is somehow limited. [...] Specifically, for DGN, it is very similar to EGNN, unless that DGN is built upon GN (with global vector u) while EGNN is upon MPNN.
>
> We agree that our architecture is similar to the EGNN proposed by Satorras et al., however we argue that the statement that the addition of the global vector u is not the only extension of the architecture. As we discuss in section G.2, the EGNN does not propagate updated edges as in a standard MPNN, whereas the DGN does propagate edge updates through each graph layer. While one may consider this a triviality, we argue that propagating the edge updates enables us to construct a more general architecture than simply EGNN + global update. Moreover, we consider a generic coordinate update rule, possibly depending only on neighboring nodes (like eq. 3, with the summation over $j\in N_I$) while in the EGNN a specific one is used, in which information is gathered from all nodes in the graph (as in Section G.2, the summation is over $j\neq i$. Using equation 3 allows one to possibly tackle local transformations, which is not possible using the update in EGNN.
>
> > 2. The proposed AGN and DGN show merits only on the simulated dataset. For the real datasets (MNIST, CIFAR10, and TSP), the performance of AGN and DGN are even worse than GN. The authors attribute this detriment to the lack of symmetry in the data. [...] Since the images for example in MNIST are actually rotation/translation equivariant, DGN is supposed to perform better than GN [...].
>
> We agree that there is apparent symmetry in the MNIST and CIFAR10 dataset when considering the data as images for a CNN to evaluate, however the embedding of the images as graphs (Dwivedi et al., 2020) causes these symmetries to be lost, as we discuss in Section 6.2. In particular, both the foreground and the background of the images are embedded as graph nodes. By doing so, the rotation of a digit in MNIST affects only a subset of nodes in the graph (since the foreground is fixed), resulting in a non-trivial transformation (certainly not belonging to the ones tackled by our architectures. We chose the TSP dataset as it is naturally described by a graphical structure, and has no apparent symmetry.
>
> > 3. In terms of evaluations on polytopes classification, all methods achieve accuracy 1 for training. Does this mean that the simulated dataset is too simple? What is the number of the training/testing samples? Is it too small?
>
> As we discuss in sections 6.1 and H.2, the training dataset consists of one sample per polytope. The test set is composed of randomly transformed versions of those in the training set, 100 per polytope. By presenting the different architectures with just one training example per polytope, we argue that we have shown that the non-equivariant architectures simply overfit to the small sample size, whereas our DGN/AGN architectures are able to learn the underlying symmetry of the polytope despite such a small training set.
>
> > 4.1 In the first paragraph, the authors state that graph neural networks, are invariant to the symmetry groups. The “symmetry group” here is too general and includes other cases besides the permutation group.
>
> This has been corrected in the revised manuscript.
>
> > 4.2 It seems AGN is only equivariant with regard to the Conformal orthogonal group (a subgroup of Conformal transformations) according to the derivations in Appendix C. So why has the claim that AGN is equivariant to the Conformal group?
>
> The AGN is invariant to the full conformal group as we show in section E, as the only requirement is that angles are preserved under a conformal transformation. As we have also focused on equivariance under the Euclidean group – which necessitates orthogonal transformations, we have also explicitly chosen to discuss the conformal orthogonal group.
>
> > 4.3 The authors are suggested to provide formal definitions of these groups: $E(n)$, $CO(R^n;Q)$ and $Conf(R^n;0)$ groups.
>
> We have chosen to not include the formal definitions of the groups, instead choosing to introduce them via derivations and examples, as a formal definition of the groups would require either an overview of Lie groups, or the assumption that the reader is already acquainted with Lie groups and representation theory. In order to make the paper accessible we have chosen to introduce the groups less formally.

---

> > ### Author Response · Authors · 2021-11-12
> > **Follow-up**
> >
> > Please feel free to ask any additional questions or make further suggestions.

---

> > ### Comment · Reviewer_ECjd · 2021-11-20
> > **Response to the authors**
> >
> > Thanks for the feedback by the authors. My concerns (particularly those related to the clarity) are partially addressed, but the major remain.
> > I appreciate the authors further discussing the difference compared to EGNN. It makes sense certain different points are considered (e.g. the global u, the edge propagation, neighboring aggregation, etc), but all are still sort of trivial and a directly enhancement upon MPNN or GraphNet, and it seems hard to convince me during the rebuttal phase unless the authors can provide something more fundamental. Besides, I still have some questions:
> >
> > 1. I accept that the symmetry of the real-world datasets are broken. But this also leads to, why not evaluate the proposed method on a symmetry-aware dataset? So we can check the effectiveness on real applications other than the synthetic experiment? Without this, we are difficult to evaluate the significance of the proposed idea.
> >
> > 2. Regarding the question 4.2 above, I am still confused. I understand that by conformal transformation, the angles are preserved (Eq. 6a), but the coordinates $x_i$ in Eq. 6d is not guaranteed to be equivariant. Does this mean that AGN is not equivariant with regard to conformal transformation?

---

> > > ### Author Response · Authors · 2021-11-22
> > > **Response to Reviewer ECjd**
> > >
> > > We thank the reviewer for the additional comments. With respect to the differences with respect to EGNN we agree on the similarities with our DGN (as we show in the appendix EGNN is a special instance of DGN). However, we would like to remark that in the paper we introduce also the AGN graph block which can deal with conformal transformations, a property that EGNN does not possess.
> > >
> > > As for the additional points:
> > > 1. We'd like to stress that we presented the polytopes classification example, the one on MNIST/CIFAR10/TSP, and QM9 in the appendix  because we believe that it's important to show both the strengths and limitations of the proposed architectures on some easy-to-visualize datasets in a pedagogical paper. However, we agree with the reviewer that a thorough benchmarking exercise against other architectures on real-world datasets is warranted; in fact, additional experiments on QM9 and other molecular datasets (like MD17) are currently a focus of investigation.
> > > 2.  As the reviewer correctly noticed Eq. 6d is not guaranteed to be equivariant. As we show in appendix E, in the AGN, the angle, edge, node and global attributes are invariant to any transformation of the coordinate embeddings that preserves the angles created by neighbouring nodes in the graph. The updated coordinates $\mathbf{x}_i^+$ can be invariant or equivariant to (some of) the same transformations depending on the particular structure of $\psi^x$ in Eq. 6d. For example, as we show in appendix C, it's possible to deal with conformal orthogonal transformations by employing Eq.3.

---

### Official Review · Reviewer_CQdG · 2021-11-05

**Correctness:** 2
**Technical Novelty And Significance:** 2
**Empirical Novelty And Significance:** Not applicable
**Recommendation:** 3
**Confidence:** 4

**Main Review:**

Pros:
  + The attempt to develop invariance and equivariant graph neural
  networks is valuable.

Concerns:

  - The proposed approach makes very strong assumptions, such as: 1)  node
    coordinates (in an ambient Euclidean space) are given; 2) the
    other noe features are  coordinate-independent.  This greatly
    limits the applicability of the proposed approach.

  -  With the above assumptions, in essence the authors explicitly "hand-code"
     specific "invariance" and "equivariance" properties, namely,    rotation/translation invariance and angle-preserving
     transformations as captured by Eucliean and conformal groups, in AGN.

 - The evaluation is done only on very limited graph datasets, namely
   the Dwivedi et al., 2020 benchmark datasets where graphs are
   generated from, e.g., 2D Euclidean datasets such as images, where
   node coordinates are well defined.


Other comments:

   - What are "global" attributes $\vec{u}$ defined on? I presume that
     they are not defined on per-node or per-edge basis. This is never
     explained.

   - Even under the assumption that node coordinates are available as
     part of the node features, you are basically assuming that the
     Euclidean distance is the right "distance" metric for the
     underlying graphs. This may not hold in general at all. For
     example, the nodes may lie in a sphere in an ambient 3-D
     Euclidean space, and the "right" distance between the nodes should be the geodesic distance on the sphere, not the ambient Euclidean distance.

 - In your model eqs.(5a) - (5b), you are making a lot of implicit
   assumptions. For example, in using 5a, you are assuming that edge
   features are independent of node coordinates, but depend only on "distances"
   between them. The same comments also apply to other equations, such
   as the transformation of the node coordinates will be based only
   node coordinates, but not other node features.

 - Without any "formal" mathematical context on how graph data are
   generated or sampled from, why would   rotation/translation/angle-preserving transformations would be the
   right "symmetry" in the data? In particular, why would preserving "angle" between neighboring (three) nodes actually capture?

 - How does your model apply to more  general    graph datasets, e.g., molecular biological networks or social
    networks such as co-authorship networks, where node coordinates
    are not available, and the whole goal of graph representation via
    node embedding is to find  "appropriate" node coordinates in a
    proper embedding space.

Minor nitpicks:
   - in your definition of equivarance, you need to first fix the group action  on Y,  $\phi'_g$, otherwise the definition does not make sense.

  - I am not sure what value the materials in Sections A-E in the appendix really add. You are basically restating the translation-invariance and angle preserving properties of Euclidean groups and conformal groups, and sections D and E hold because you explicitly construct AGA to be distance-invariant and angle preserving.

**Summary Of The Paper:**

The paper seeks to develop invariance and equivariant graph neural networks.
Based on the assumption that node feautures can be separated into a
node coordinate feature vector and coordinate-independent node feature
vector, the authors develop a GNN architecture, called "angle
preserving graph network" (AGN), that is  equivariant to  the
Euclidean group and conformal group. The method is evaluated using the
"benchmark" graph datasets produced in (Dwivedi et al., 2020) from,
e.g., MNIST and CIFAR10 datasets.

**Summary Of The Review:**

The paper seeks to develop invariance and equivariant graph neural
 networks. The proposed approach makes strong assumptions. As such, it
 has very limited applicability. The evaluation is also very limited.

---

> ### Author Response · Authors · 2021-11-09
> **Rebuttal to Reviewer CQdG**
>
> We thank the reviewer for the provided comments and we address them below.
>
> > - The proposed approach makes very strong assumptions, such as: 1) node coordinates (in an ambient Euclidean space) are given; 2) the other node features are coordinate-independent. [...]
>
> We presented our methods by considering graphs embedded in a Euclidean space where invariance and equivariance to Euclidean and/or conformal transformations are easy to visualize. However, as we state in the paper, this assumption can be relaxed and one can apply them in a different context by properly defining node coordinates (provided that one is looking for invariance or equivariance on them).
>
> > - With the above assumptions, in essence the authors explicitly "hand-code" specific "invariance" and "equivariance" properties [...]
>
> That's exactly the point of our work and many other ones (see the Related work section): to build better/stronger inductive biases in the architectures themselves to guide and make the learning process easier or more efficient. Note that even CNNs are built upon this concept through the convolution operator.
>
> > - The evaluation is done only on very limited graph datasets, namely the Dwivedi et al., 2020 benchmark datasets [...] where node coordinates are well defined.
>
> As we have said, node coordinates must be defined in order for the methods to be applied. In addition to MNIST, CIFAR10 and TSP datasets we consider a polytope classification problem and QM9. The polytopes example provides a comparison between differing architectures, and enables one to understand the importance of symmetries on such a dataset. The paper is pedagogical and mostly a theoretical introduction to our method: working on applications to other datasets is the subject of ongoing research.
>
> > - What are "global" attributes $\mathbf{u}$ defined on? I presume that they are not defined on per-node or per-edge basis. This is never explained.
>
> Global attributes are attributes of the whole graph. They can be an important component and are widely used in modern GNN-based architectures (see e.g. Battaglia et al, 2018, https://arxiv.org/abs/1806.01261).
>
> > - Even under the assumption that node coordinates are available as part of the node features, you are basically assuming that the Euclidean distance is the right "distance" metric for the underlying graphs. This may not hold in general at all. For example, the nodes may lie in a sphere in an ambient 3-D Euclidean space, and the "right" distance between the nodes should be the geodesic distance on the sphere, not the ambient Euclidean distance.
>
> That's an interesting point but it's a different problem with respect to the one addressed in this paper.
>
> > - In your model eqs.(5a) - (5b), you are making a lot of implicit assumptions. For example, in using 5a, you are assuming that edge features are independent of node coordinates, but depend only on "distances" between them. The same comments also apply to other equations, such as the transformation of the node coordinates will be based only node coordinates, but not other node features.
>
> Yes, as we specify in the paper and in sections D and E in the appendix, those are the architectural choices that allow the proposed graph blocks to be invariant/equivariant to the considered transformations.
>
> > - Without any "formal" mathematical context on how graph data are generated or sampled from, why would rotation/translation/angle-preserving transformations would be the right "symmetry" in the data? In particular, why would preserving "angle" between neighboring (three) nodes actually capture?
>
> If we correctly understand the question, this is the point we address in Section 6.2 by showing that if the expected symmetries are not in the data, using this type of architectures is counterproductive.
>
> > - How does your model apply to more general graph datasets, e.g., molecular biological networks or social networks such as co-authorship networks, where node coordinates are not available, and the whole goal of graph representation via node embedding is to find "appropriate" node coordinates in a proper embedding space.
>
> In general it doesn't, at least in a meaningful way. In fact, what would be the value or meaning of being invariant to Euclidean or conformal transformations in a co-authorship network?
>
> > - in your definition of equivarance, you need to first fix the group action on Y, $\varphi_g'$, otherwise the definition does not make sense.
>
> As per the definition of equivariance we give (section 2.1) the group action on Y is already fixed.
>
> > - I am not sure what value the materials in Sections A-E in the appendix really add [...]
>
> Sections A-E are meant to help the reader to better understand and have more details (and proofs) on the architectural choices we made in the main paper. In particular, section C provides a number of possible choices for the coordinate update functions to be used in equation 5c and 6d.

---

> > ### Author Response · Authors · 2021-11-12
> > **Follow-up**
> >
> > Please feel free to ask any additional questions or make further suggestions.

---

### Decision · Program_Chairs · 2022-01-20

**Decision:**

Reject

**Comment:**

This work proposes to extend the invariance/equivariance properties of GNNs by focusing on distance-preserving and angle-preserving transformations, given respectively by the Euclidean and Conformal group. Preliminary experiments are reported that demonstrate the advantage of such architectures.
Reviewers found this work generally interesting, tackling an important problem and proposing a valid solution. However, they also raised important concerns, namely the relatively minor novelty relative to recent models (such as EGNN), as well as the lack of convincing real-world experiments that would validate the modeling assumptions. Taking all these considerations into account, the AC recommends rejection at this time, and encourages the authors to address the points raised by reviewers in a revision.